# Bilevel Optimization with Lower-Level Contextual MDPs

## Abstract

Recent research has focused on providing the right incentives to learning agents in dynamic settings. Given the high-stakes applications, the design of reliable and trustworthy algorithms for these problems is paramount. In this work, we define the Bilevel Optimization on Contextual Markov Decision Processes (BO-CMDP) framework, which captures a wide range of problems such as dynamic mechanism design or principal-agent reward shaping. BO-CMDP can be viewed as a Stackelberg Game where the leader and a random context beyond the leader's control together configure an MDP while (potentially many) followers optimize their strategies given the setting. To solve it, we propose Hyper Policy Gradient Descent (HPGD) and prove its non-asymptotic convergence. We make very weak assumptions about the information available. HPGD does not make any assumption about competition or cooperation between the agents and allows the follower to use any training procedure of which the leader is agnostic. This setting aligns with the information asymmetry present in most economic applications.

The Markov Decision Process (MDP) (Puterman, 2014) is a versatile framework to model sequential decision-making problems in health care (Yu et al., 2021), energy systems (Perera & Kamalaruban, 2021), economics (Charpentier et al., 2021), and finance (Hambly et al., 2023) among many others domains. Much work exists on finding optimal policies for a given MDP (Sutton & Barto, 2018). However, in many applications, an MDP can be configured on purpose or affected by exogenous events, both of which can significantly alter the optimal decision-making policies.

Consider for instance a macroeconomic model in which

[1]Anonymous Institution, Anonymous City, Anonymous Region, Anonymous Country. Correspondence to: Anonymous Author <anon.email@domain.com>.

Submitted to ICML Agentic Markets Workshop. Do not distribute.

households optimize their consumption and resource allocation to maximize utility. Their optimal behavior depends on exogenous variables such as macroeconomic trends, prices, or geopolitical events that are beyond the control of any participant in the system. Value-added and income tax rates, however, can be configured and optimized by a central authority optimizing the system's overall welfare. We formalize several important economic problems, such as dynamic mechanism design, tax design for macroeconomic modeling, a dynamic principal-agent problem, as well as meta RL in our problem formulation in Appendix A.

In this work, we address how to optimize configurations for a contextual MDP when some parameters are configurable while others are stochastic. We propose the Bilevel Optimization on Contextual Markov Decision Processes (BO-CMDP) framework that generalizes many previous models including Configurable MDPs (Metelli et al., 2018), contextual bilevel optimization (Hu et al., 2024), adaptive model design (Zhang et al., 2018; Chen et al., 2022), and Meta-RL (Beck et al., 2023), and finds many applications in the dynamic Stakelberg Games (Gerstgrasser & Parkes, 2023; Wang et al., 2023), Security Games (Sinha et al., 2018; Letchford & Vorobeychik, 2013), dynamic mechanism design (Curry et al., 2024), and economics (Curry et al., 2023; Zheng et al., 2022; Hill et al., 2021). We discuss related works in detail in Appendix B.

To solve BO-CMDP, we propose Hyper Policy Gradient Descent a stochastic bilevel optimization algorithm that *is agnostic of the learning dynamics of the agent*. We establish the non-asymptotic convergence rate of our algorithm to a stationary point of the overall objective. Additionally, we demonstrate the performance of HPGD in a grid-world design problem and showcase that in most cases it matches the performance of benchmark algorithms with stronger assumptions and for certain parameters it outperforms them.

## 1. Problem Formulation

We consider a bilevel optimization problem, where the followers solve Contextual Markov Decision Processes (CM-PDs) and the leader (partially) controls the configuration of the CMDPs. In particular, the leader chooses a parameter $x \in X \subseteq \mathbb{R}^d$ and nature chooses a random con-

text $\xi$ according to a distribution $\mathbb{P}_\xi$. Together $(x, \xi)$ parameterizes an MDP $\mathcal{M}_{x,\xi}$, which the follower aims to solve. $\mathcal{M}_{x,\xi}$ is defined by a tuple $(\mathcal{S}, \mathcal{A}, r_{x,\xi}, P_{x,\xi}, \mu_{x,\xi}, \gamma)$, where $\mathcal{S}$ denotes the state space, $\mathcal{A}$ denotes the action space, $r_{x,\xi}(\cdot, \cdot) : \mathcal{S} \times \mathcal{A} \to \mathbb{R}$ is the reward function, $P_{x,\xi}(\cdot; \cdot, \cdot) : \mathcal{S} \times \mathcal{S} \times \mathcal{A} \to [0, 1]$ denotes the transition kernel, $\mu_{x,\xi}$ indicates the initial state distribution, and $\gamma$ is the discount factor. The subscript $x, \xi$ implies that rewards, transitions, and initial state distribution depend on the leader's decision $x$ and the context $\xi$. Connecting to previous works, for a fixed $x$, $\mathcal{M}_{x,\xi}$ is a *contextual MDP* (Hallak et al., 2015) with respect to $\xi$. For a fixed $\xi$, $\mathcal{M}_{x,\xi}$ generalizes a *configurable MDP* (Metelli et al., 2018). Given $\mathcal{M}_{x,\xi}$, the follower maximizes an entropy-regularized objective by choosing a policy $\pi_{x,\xi}$, where $\pi_{x,\xi}(a; s)$ denotes the probability of choosing action $a$ in state $s$.

$$\max_\pi J_{\lambda,x,\xi}(\pi) = \mathbb{E}_{s_0} \left[ V_{\lambda,x,\xi}^\pi(s) \right]$$

$$= \mathbb{E}_{s_0} \left[ \mathbb{E}_{P_{x,\xi}}^\pi \left[ \sum_{t=0}^\infty \gamma^t \left( r_{x,\xi}(s_t, a_t) + \lambda H(\pi; s_t) \right) \right] \right] \quad (1)$$

where $s_0 \sim \mu_{x,\xi}$, $a_t \sim \pi(\cdot; s_t)$, $s_{t+1} \sim P_{x,\xi}(\cdot; s_t, a_t)$ and $H(\pi; s) = \sum_a \pi(a; s) \log \pi(a; s)$. We call $\lambda \geq 0$ the regularization parameter and $V_{\lambda,x,\xi}^\pi$ the value function. As standard in RL literature, we define the related Q and advantage functions as:

$$Q_{\lambda,x,\xi}^\pi(s, a) = r_{x,\xi}(s, a) + \gamma \mathbb{E}_{s' \sim P_{x,\xi}(\cdot; s,a)} \left[ V_{\lambda,x,\xi}^\pi(s') \right]$$

$$A_{\lambda,x,\xi}^\pi(s, a) = Q_{\lambda,x,\xi}^\pi(s, a) - \sum_{a'} \pi(a'; s) Q_{\lambda,x,\xi}^\pi(s, a').$$

$$(2)$$

The unique optimal policy for (1) is denoted by $\pi_{x,\xi}^*(s; a) \propto \exp(Q_{\lambda,x,\xi}^*(s, a)/\lambda)$, i.e., the softmax of the optimal Q-function (Nachum et al., 2017).[1] Given $x$, $\pi_{x,\xi}^*$ and $\xi$, the leader in turn incurs a loss $f(x, \pi_{x,\xi}, \xi) \in \mathbb{R}$, which it wants to minimize in expectation over $\mathbb{P}_\xi$. BO-CMDP can thus be formulated as the following stochastic bilevel optimization.

$$\min_x \quad F(x) := \mathbb{E}_\xi[f(x, \pi_{x,\xi}^*, \xi)] \quad \text{(leader, upper-level)}$$

$$\text{where} \quad \pi_{x,\xi}^* = \underset{\pi}{\arg\max} \, J_{\lambda,x,\xi}(\pi). \quad \text{(follower, lower-level)}$$

$$(3)$$

Equation (3) is well-defined due to entropy regularization, which ensures the uniqueness of $\pi_{x,\xi}^*$. Entropy-regularization also turns $\pi_{x,\xi}^*$ differentiable, often stabilizes learning and appears in previous works (Chen et al., 2022). Moreover, the difference between the entropy-regularized and unregularized problem generally vanishes as $\lambda$ goes to 0 (Chen et al., 2022; Dai et al., 2018; Geist et al., 2019).

---

[1]For brevity, we notationally drop the dependence of $\pi_{x,\xi}$ on $\lambda$, but keep it for $V_{\lambda,x,\xi}^\pi$ to emphasize the entropy-regularization.

## 2. Hyper Policy Gradient Descent Algorithm for BO-CMDP

In this section, we derive a simple expression for the hypergradient of BO-CMDP. We present HPGD and prove non-asymptotic convergence. We show this is the case for several popular RL algorithms. In Appendix C, we present further results for two important special cases of our problem: (1) when the upper-level objective decomposes as a discounted sum of rewards over the lower-level trajectories, and (2) when the leader can direct the lower-level algorithm. The proofs of the results in this Section are deferred to Appendix E. We make the following standard assumptions on how $x$ and $\xi$ influence the setup of the CMDP.

**Assumption 2.1.** We assume that $f$ is $L_f$-Lipschitz and $S_f$-smooth in $x$ and $\pi$, uniformly for all $\xi$ and that $\forall x, \xi : |r_{x,\xi}(s, a)| < \overline{R}$, $\|\partial_x \log P_{x,\xi}(s'; s, a)\|_\infty < K_1$, $\|\partial_x r_{x,\xi}(s, a)\|_\infty < K_2$.

### 2.1. Hypergradient derivation

The leader's loss $f$ depends on both $x$ and the optimal policy $\pi_{x,\xi}^*$. Therefore, the derivative of $f$ with respect to $x$ is commonly referred to as the *hypergradient* to highlight this nested dependency. Using the implicit function theorem (Ghadimi & Wang, 2018), we obtain a closed-form expression of the hypergradient. However, it involves computing and inverting the Hessian of the follower's value function, which can be computationally expensive and unstable (Fiez et al., 2020; Liu et al., 2022). Instead, we leverage the fact that the formulation of $\pi_{x,\xi}^*$ is a softmax function and explicitly compute its derivative with respect to $x$. Applying the Dominated Convergence Theorem to switch derivative and expectation, we arrive at Theorem 2.2.

**Theorem 2.2.** *Under Assumption 2.1, F is differentiable and the hypergradient is given by*

$$\frac{dF(x)}{dx} = \mathbb{E}_\xi \left[ \frac{\partial_1 f(x, \pi_{x,\xi}^*, \xi)}{\partial x} \right.$$

$$\left. + \mathbb{E}_{s \sim \nu, a \sim \pi_{x,\xi}^*} \left[ \frac{1}{\lambda \nu(s)} \frac{\partial_2 f(x, \pi_{x,\xi}^*, \xi)}{\partial \pi_{x,\xi}^*(a; s)} \partial_x A_{\lambda,x,\xi}^{\pi_{x,\xi}^*}(s, a) \right] \right]$$

$$(4)$$

*where $\nu$ is any sampling distribution with full support on the state space $\mathcal{S}$.*

The first term captures the direct influence of $x$ on $f$, and the second is the indirect influence through $\pi_{x,\xi}^*$. We assume the leader knows $\partial_1 f(\cdot, \pi, \xi)$ and $\partial_2 f(x, \cdot, \xi)$. To compute $\partial_x A_{\lambda,x,\xi}^{\pi_{x,\xi}^*}(s, a)$, i.e. the partial derivative with respect to $x$ for a fixed policy, we need to know $\partial_x Q_{\lambda,x,\xi}^{\pi_{x,\xi}^*}(s, a)$ (cf. (2)). We derive an expression for the latter in Theorem 2.3. The proof adapts the analysis of the policy gradient theorem to account for the dependence of $P_{x,\xi}$, $\mu_{x,\xi}$ and $r_{x,\xi}$ on $x$.

**Algorithm 1** Hyper Policy Gradient Descent (HPGD)

    **Input:** Iterations $T$, Learning rate $\alpha$, Regularization $\lambda$,
    Trajectory oracle $o$, Initial point $x_0$
    **for** $t = 0$ to $T - 1$ **do**
        $\xi \sim \mathbb{P}_\xi$, $s \sim \nu$ and $a \sim \pi_{x,\xi}^o(\cdot; s)$
        $\widehat{\partial_x A_{\lambda,x,\xi}^{\pi_{x,\xi}^o}}(s,a) \leftarrow \texttt{GradEst}(\xi, x_t, s, a, o)$ (Alg. 2)
        $\widehat{\frac{dF}{dx}} \leftarrow \frac{\partial_1 f(x_t, \pi_{x_t,\xi}^o, \xi)}{\partial x} + \frac{\partial_2 f(x_t, \pi_{x_t,\xi}^o, \xi)}{\partial \pi(s,a)} \frac{\widehat{\partial_x A_{\lambda,x,\xi}^{\pi_{x,\xi}^o}}(s,a)}{\lambda \nu(s)}$
        $x_{t+1} \leftarrow x_t - \alpha \widehat{\frac{dF}{dx}}$
    **end for**
    **Output:** $\hat{x}_T \sim \text{Uniform}(\{x_0, \ldots, x_{T-1}\})$

**Theorem 2.3.** *For given $\pi, x, \xi$, it holds that:*

$$\partial_x Q_{\lambda,x,\xi}^\pi(s,a) = \mathbb{E}_{s,a}^\pi \left[ \sum_{t=0}^\infty \gamma^t \frac{dr_{x,\xi}(s_t, a_t)}{dx} \right.$$
$$\left. + \gamma^{t+1} \frac{d \log P_{x,\xi}(s_{t+1}; s_t, a_t)}{dx} V_{\lambda,x,\xi}^\pi(s_{t+1}) \right].$$

Note, Theorems 2.2 and 2.3 generalize existing results in model design for MDPs to CMDPs (Chen et al., 2022; Zhang et al., 2018).

## 2.2. HPGD Algorithm and Convergence Analysis

To minimize $F(x)$, one would ideally sample unbiased estimates of the hypergradient in Equation (4) and run stochastic gradient descent (SGD). However, the leader does not have access to $\pi_{x,\xi}^*$ and generally no control over the training procedure of the lower level. Instead, we assume the follower adapts any preferred algorithms to solve the MDP up to a certain precision $\delta$ and the leader can only observe trajectories from the follower's policy, as motivated by several practical applications.

**Assumption 2.4.** *For any $\mathcal{M}_{x,\xi}$, the leader has access to an oracle $o$, which returns trajectories sampled from a policy $\pi_{x,\xi}^o$ such that $\forall x, \forall \xi : \mathbb{E}_o \left[ \left\| \pi_{x,\xi}^* - \pi_{x,\xi}^o \right\|_\infty^2 \right] \leq \delta^2$.*

We will show that Assumption 2.4 is relatively mild and holds for a variety of RL algorithms. Given access to trajectories generated by $\pi_{x,\xi}^o$, the leader can construct an estimator of $\partial_x A_{\lambda,x,\xi}^{\pi_{x,\xi}^o}(s,a)$ by rolling out $\pi_{x,\xi}^o$ for $T$ steps, where $T \sim \text{Geo}(1 - \gamma)$. We defer the construction (Algorithm 2) and proof of unbiasedness (Proposition E.2) to the Appendix. Using this estimator, we introduce HPGD in Algorithm 1. As $F$ is generally nonconvex due to the bilevel structure (Ghadimi & Wang, 2018), we demonstrate non-asymptotic convergence to a stationary point of $F$, which matches the lower bound for solving stochastic smooth nonconvex optimization (Arjevani et al., 2023).

**Theorem 2.5.** *Under Assumption 2.1 and Assumption 2.4, we have the following result for HPGD:*

$$\mathbb{E} \left\| \frac{dF(\hat{x}_T)}{dx} \right\|^2 = \mathcal{O}\left( \frac{1}{\alpha T} + \delta + \alpha \right). \tag{5}$$

*For $\alpha = \mathcal{O}(1/\sqrt{T})$ and $\delta = \mathcal{O}(1/\sqrt{T})$, HPGD converges to a stationary point at rate $\mathcal{O}(1/\sqrt{T})$.*

*Proof sketch.* Using the smoothness of $F$ and the fact that $\hat{x}_T$ is uniformly sampled from all iterates, we upper bound the left side of (5) by the sum of three terms. The first is $|F(x_0) - \min_x F(x)|/\alpha T$. The second depends on the bias of our gradient estimate, which we show is linear in $\delta$. The last term depends on $\alpha$ times the variance of our estimator, which is bounded. $\square$

A major advantage of HPGD is that the follower can use a multitude of algorithms to solve the lower-level MDP, while the leader only needs access to generated trajectories. While Assumption 2.4 certainly holds if the follower solves the MDP exactly, for example with an LP-solver, we are interested in verifying Assumption 2.4 for common RL algorithms, which can scale to larger state and action spaces. In Appendix E, we prove non-asymptotic convergence to $\pi_{x,\xi}^*$ for Value Iteration, which converges at rate $\mathcal{O}(\log 1/\delta)$ (Proposition E.4); Q-learning, which converges at rate of $\mathcal{O}(\log(1/\delta)/\delta^2)$ (Proposition E.5) and Natural Policy Gradient, which converges at rate of $\mathcal{O}(\log 1/\delta)$ (Proposition E.7). Additionally, we show Vanilla Policy Gradient converges asymptotically in Proposition E.6. All these Algorithms thus satisfy Assumption 2.4, which makes HPGD widely applicable and the followers might use a variety of model-free or model-based algorithms.

## 3. Numerical Experiments

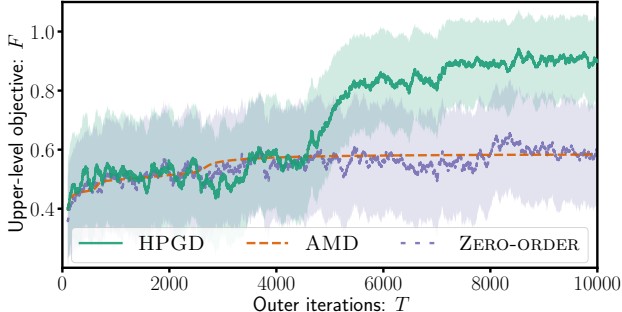

*Figure 1.* Upper-level objective values, $F$, over the number of outer iterations. HPGD escapes local optima achieving higher performance than comparison algorithms.

We illustrate the performance of HPGD in the Four-Rooms environment and compare it to Adaptive Model Design

(AMD) (Chen et al., 2022) and a zeroth-order gradient approximation algorithm. We describe the algorithms in Appendix F.1.1 and Appendix F.1.2. Technical details about the implementation are deferred to Appendix (F.2).

The Four-Rooms environment consists of a grid world divided into 4 rooms as shown at Figure 2. $S$ denotes the initial position while $G^1$ and $G^2$ are goal states. We consider the two goal states as separate tasks and define $\xi$ in Equation (3) to be the uniform distribution over the set of tasks, i.e., $\xi \sim \mathrm{Uniform}(\{1,2\})$. We denote the goal state in each task by $G^\xi$. The state space $\mathcal{S}$ is defined by the cells of the grid world while the actions are the movements in the four directions. In each step $t$, with probability $2/3$, the agent moves to $s_{t+1}$ following the chosen direction $a_t$ while it takes a random movement with probability $1/3$. The reward is always zero except when $s_t = G^\xi$ where $r(s_t, a_t) = 1$, and the episode resets. To incentivize taking the shortest path, we set the discount factor as $\gamma = 0.99$.

For the upper level, we let $x$ parameterize an additive penalty function $\tilde{r}_x : \mathcal{S} \times \mathcal{A} \to [-0.2, 0.0]$ [2], such that the follower receives a reward of $r + \tilde{r}_x$, as in the principal-agent problem (Ben-Porat et al., 2024). The goal of the leader is to steer the followers through the cell marked with $+1$ in Figure 2, denoted by $s^{+1}$, while keeping the penalties allocated to states to their minimum. We define $\overline{r}$ in Equation (7) as

$$\overline{r}_{x,\xi}(s_t, a_t) = \mathbb{I}_{\{s_t = s^{+1}\}} - \beta \mathbb{I}_{\{s_t = G^\xi\}} \sum_{s,a} \tilde{r}_x(s, a),$$

where $\mathbb{I}$ is the indicator function and the second term defines the cost associated with implementing the penalties for the lower level. Note that there is a trade-off between the terms in $\overline{r}$ depending on the context variable $\xi$. If $\xi = 2$, the desired change in the follower's policy can be achieved with small interventions since the shortest path from $S$ to $G^2$ is already going through the bottom-left room. When $\xi = 1$, the leader must completely block the shortest path from $S$ to $G^1$ to divert the follower through the desired state. An efficient algorithm for this BO-CMDP problem therefore must avoid the local optimum of setting $\tilde{r} = 0$ and find the balance between the follower visiting state $s^{+1}$ and implementing large quantity of penalties in the CMDP.

Figure (1) depicts the upper-level's objective function over the learning iterations $t$ with hyperparameters $\lambda = 0.001$ and $\beta = 1.0$. HPGD outperforms both AMD and the zero-order algorithms in this instance in terms of overall performance. The major difference in their performances is that HPGD successfully escapes the local optimum of $\tilde{r} = 0$ after about 5000 steps and assigns all the additive penalty budget to states in the grid world. On the contrary, AMD and zero-order converge to the local optimum of minimizing the implementation penalty term in $\overline{r}$.

---

[2]The parametrization of this function is described in Appendix F.2.1.

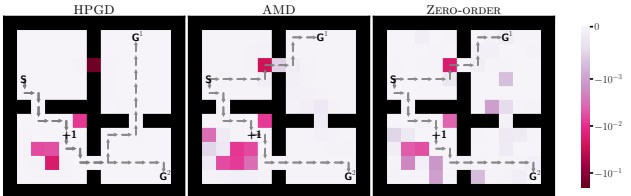

Figure 2. Reward penalties given to the lower-level agent in each state of the Four-Rooms problem optimized by the HPGD, AMD, and zero-order, respectively. HPGD efficiently steers the lower-level MDP when the task is to reach $G^1$ while others are only successful in the case of $G^2$.

Figure 2 shows the value of additive penalties $\tilde{r}$ in the state space with the highest probability paths for the goal states. HPGD successfully blocks the follower when $\xi = 1$ and diverts its shortest path from $S$ to $G^1$ along the other rooms, while AMD and zero-order fail to assign sufficient penalty to the upper corridor to cause the same effect. All algorithms are successful in ensuring that the shortest path through the bottom-left room is going through the marked state.

The parameters $\lambda$ and $\beta$ were chosen for demonstration purposes to highlight the capability of HPGD to escape local minima, as has been observed for SGD (Xie et al., 2021). However, we emphasize that in the majority of the cases, the three algorithms perform equally as shown in Table 1 in Appendix F.2.3. We provide the figures for the remaining hyperparameters in Appendix F.2.4. The slightly higher performance of AMD and low standard error among initializations is expected since this algorithm calculates the gradient of $f$ deterministically while HPGD and zero-order rely on stochastic estimates yielding more variations, especially for the zero-order approach.

## 4. Conclusion

We introduce the class of bilevel optimization problems with lower-level contextual MDPs that capture a wide range of important applications, in particular in economics. We propose an oracle-based algorithmic framework HPGD and analyze its convergence, as well as sample complexities. Importantly, HPGD works with any existing algorithm that solves the lower-level MDP to near-optimality, making it suitable in various regimes when the leader can only observe trajectories of the follower. Numerical results further validate the expressiveness of BO-CMDP and the performance of HPGD. Future directions include algorithm design and exploring the sample complexity and variance tradeoff when the leader can fully control the followers' training, as well as deploying HPGD to larger settings from the described application areas.

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

## A. Applications formalized as Bilevel Optimization on Contextual Markov Decision Processes

**Dynamic Mechanism Design** considers the problem of a mechanism designer controlling an MDP. Bidders have no control over the MDP but have a type $\xi$, distributed according to some prior distribution, which parameterizes their reward functions. Based on their rewards, they bid on the trajectories of the MDP. Curry et al. (2024) consider the setting where the mechanism designer wants to elicit truthful bids but also maximize some other objective $-\mathcal{L}$, such as revenue. They restrict to dynamic affine maximizers, where the mechanism designer first chooses a set of agent-dependent weights $x_{w,i}$ and state-action dependent boosts $x_b$ to affinely transform social welfare and then learns a policy to maximize this affine social welfare. This problem formulation can be exactly captured as BO-CMDP, as follows:

$$\min_x \mathbb{E}_\xi[\mathcal{L}\left(\pi^*_{\xi,x_w,x_b}, x_w, x_b\right)] \text{ s.t. } \pi^*_{\xi,x_w,x_b} = \arg\max_\pi \mathbb{E}_{s_t,a_t\sim\pi}\left[\sum_{t=0}^{T}\left(\sum_{i=1}^{n} x_{w,i}r_i(s_t,a_t)\right) + x_b(s_t,a_t)\right] \tag{6}$$

**Tax Design for Macroeconomic Modeling** consider a public entity setting tax rates and representative households responding optimally by balancing their short-term utility of consumption and long-term wealth accumulation (Hill et al., 2021; Chen et al., 2022; Zheng et al., 2022). A potential formulation of this problem as a BO-CMDP is

$$\max_{x,y} \mathbb{E}_\xi\left[\phi(x,y,\pi^*_{x,y,\xi},\xi)\right] \text{ s.t. } \pi^*_{x,y,\xi}(\cdot) = \underset{\pi}{\text{argmax}} \mathbb{E}\left[\sum_{t=0}^{\infty}\gamma^t\left(r^W_\xi(s_t) + r^C_{x,\xi}(\pi(s_t))\right)\right],$$

where $\phi$ defines the social welfare objective of the leader. The state $s_t$ defines the wealth of a household while their actions decide their working hours and consumption in each time step. The reward function $r^W_\xi$ and $r^C_{x,\xi}$ define the households' utility functions for wealth and consumption, respectively. The value-added tax rate $x$ affects the consumption utility function $r^C_{x,\xi}$ while the income tax $y$ changes the transition kernel modeling wealth accumulation. $\xi$ represents the preferences of the households over several consumption goods and their productivity in this problem formulation.

**Population Principal-Agent Reward Shaping** considers a principal aiming to craft a non-negative bonus reward function $r^B_x$, parameterized by $x$, to motivate an agent (Ben-Porat et al., 2024; Yu & Ho, 2022; Zhang & Parkes, 2008). Commonly, a principal faces multiple agents that form a distribution. Each agent has its own individual reward function $r_\xi$. This scenario, termed *population principal-agent reward shaping* is captured by our BO-CMDP framework.

$$\max_x \mathbb{E}_\xi\left[\sum_{t=0}^{\infty}\gamma^t\overline{r}(s_t,\pi^*_{x,\xi}(s_t))\right] \text{ s.t. } \pi^*_{x,\xi}(\cdot) = \underset{\pi}{\text{argmax}} \mathbb{E}\left[\sum_{t=0}^{\infty}\gamma^t\left(r_\xi(s_t,\pi(s_t)) + r^B_x(s_t,\pi(s_t))\right)\right].$$

Here $\mathbb{E}_\xi$ denotes the expectation over the distribution of agents and the trajectories. The policy $\pi^*_{x,\xi}(\cdot)$ is the optimal response of the $\xi$-th agent to the composite reward function $r_\xi + r^B_x$. The principal's reward is $\overline{r}(s_t,a_t)$ when the agent visits the state action pair $(s_t,a_t)$.

**Meta reinforcement learning (Meta RL)** aims to leverage the similarity of several RL tasks to learn common knowledge and use it on new unseen tasks (Beck et al., 2023). One way to formulate Meta RL problems is to find a common regularization policy $\tilde{\pi}$ for multiple tasks.

$$\max_{\tilde{\pi}} \mathbb{E}_\xi\left[\sum_{t=0}^{\infty}\gamma^t r_\xi(s_t,\pi^*_{\tilde{\pi},\xi}(s_t))\right] \text{ s.t. } \pi^*_{\tilde{\pi},\xi}(\cdot) = \underset{\pi}{\text{argmax}}\left[\sum_{t=0}^{\infty}\gamma^t r_\xi(s_t,\pi(s_t)) - \frac{\lambda}{2}KL(\pi(s_t)||\tilde{\pi}(s_t))\right],$$

where $\xi$ represents the distribution of multiple RL tasks and $r_\xi$ is the reward for the task indexed by $\xi$.

Note, that previous works in these areas have either focused on the setting with a single representative follower (Ben-Porat et al., 2024; Chen et al., 2022) or presented a problem-specific algorithm that cannot capture our BO-CMDP framework in its full generality (Beck et al., 2023; Ben-Porat et al., 2024; Curry et al., 2024).

## B. Related Works

**Related Work**

**Stochastic bilevel optimization** has been extensively explored in the literature (Dempe, 2002; Bard, 2013). In recent years, there is a pivotal shift to non-asymptotic analysis of stochastic gradient methods (Ghadimi & Wang, 2018; Chen et al., 2021;

Khanduri et al., 2021; Kwon et al., 2023; 2024). (Hu et al., 2024) propose contextual stochastic bilevel optimization where the lower level solves a static contextual optimization. Our work generalizes to the lower level solving a contextual MDP. This poses unique challenges in terms of hypergradient estimation and sample generation. Leveraging the special structure of BO-CMDP, we avoid Hessian and Jacobian estimation of the lower-level MDP when computing the hyper policy gradient, which is crucial for scalability.

**Configurable MDP** (ConfMDP (Metelli et al., 2018)) is an extension of a traditional MDP allowing external parameters or settings to be adjusted by the decision-maker, often referred to as the *configurator*. Only recently some works studied the case where the configurator has a different objective than the agent (Ramponi et al., 2021). However, that work assumes access to a finite number of parameters that the configurator can control, while our model goes beyond this assumption. In addition, our model captures the variability and uncertainty that the agent could face in the same configuration environment.

**Steering RL agents** considers how to design additional rewards or otherwise change the MDP to observe desirable learning outcomes. There exist several work strands in this area, such as environment design for generalization (Dennis et al., 2020; Diaz et al., 2022; Yang et al., 2022), reward shaping (Hadfield-Menell et al., 2017; Hu et al., 2020) and model design (Chen et al., 2022; Zhang et al., 2018). In this work, we capture the problem settings of the latter two works as a special case, where the context is trivial, the algorithm becomes deterministic and the leader can either only influence the transition probabilities or has direct access to the learning dynamics of the follower. For general no-regret learners Zhang et al. (2024) present several theoretical results.

**Stackelberg games** are a game theoretic framework, where a leader takes actions to which one or multiple followers choose the best response (Stackelberg, 1934). Several existing lines of work have studied solving variants of Stackelberg games. Examples include Stackelberg equilibrium solvers (Fiez et al., 2020; Gerstgrasser & Parkes, 2023), opponent shaping (Foerster et al., 2018; Yang et al., 2020), mathematical programs with equilibrium constraints (Liu et al., 2022; Wang et al., 2023; 2022; Zhang et al., 2023), inducing cooperation (Baumann et al., 2020; Balaguer et al., 2022) and steering economic simulations (Curry et al., 2023; Zheng et al., 2022). These works are either kept general with limited implications for our problem or consider entirely distinct settings.

Moreover, as outlined in Appendix A BO-CMDP exactly captures several existing practical problems, such as optimal dynamic mechanism design (Curry et al., 2024), Principal-Agent problems (Ben-Porat et al., 2024), and Meta RL (Beck et al., 2023). This highlights the practical relevance and impact of our proposed algorithms.

## C. Additional Theoretical results

Here we present additional results for two special cases:(1) when the upper-level objective decomposes as a discounted sum of rewards over the lower-level trajectories, and (2) when the leader can direct the lower-level algorithm.

### C.1. Upper-Level Discounted Reward Objective

So far we have assumed the leader knows $\partial_1 f(\cdot, \pi, \xi)$ and $\partial_2 f(x, \cdot, \xi)$. In this section, instead, we assume $f$ can be written as the negative expected sum of discounted rewards over the lower-level trajectories and show how to compute the hypergradient. In many practical applications, such as reward shaping, meta RL (cf. **??**), or dynamic mechanism design (Curry et al., 2024), the loss $f$ satisfies:

$$f(x, \pi^*_{x;\xi}, \xi) = -\mathbb{E}^{\pi^*_{x,\xi}}_{s_0 \sim \mu}\Big[ \sum_t \gamma^t \overline{r}_{x,\xi}(s_t, a_t)\Big]. \tag{7}$$

Here $\overline{r}_{x,\xi}$ represents the reward of the leader, which is generally distinct from the follower's reward. The expectation is taken over trajectories induced by the lower-level $\pi^*_{x,\xi}$. In this case, the leader does not know the partial derivatives of $f$ but can estimate them from trajectory samples.

**Proposition C.1.** *If $f$ decomposes as in Equation* (7), *then $\frac{dF(x)}{dx}$ can be expressed as follows:*

$$\frac{dF(x)}{dx} = \mathbb{E}_\xi\Bigg[\mathbb{E}^{\pi^*_{x,\xi}}_{s_0 \sim \mu_{x,\xi}}\Bigg[ \sum_{t=0}^{\infty} \gamma^t \Bigg( \frac{1}{\lambda}\partial_x A^{\pi^*_{x,\xi}}_{\lambda,x,\xi}(s_t, a_t)\overline{Q}_{x,\xi}(s_t, a_t)$$

$$+ \frac{d\overline{r}_{x,\xi}(s_t, a_t)}{dx} + \partial_x \log P_{x,\xi}(s_t; s_{t-1}, a_{t-1})\overline{V}_{x,\xi}(s_t) \Bigg) \Bigg]\Bigg], \tag{8}$$

*where for compactness, we slightly abuse notation to express $\mu_{x,\xi}(s_0)$ as $P_{x,\xi}(s_0, a_{-1}, s_{-1})$.*

Here $\overline{V}_{x,\xi}, \overline{Q}_{x,\xi}$ are the (unreqgularized) value and Q-functions with respect to $\overline{r}_{x,\xi}$. Comparing to Theorem 2.2, note that here the expectation is over trajectories with starting states distributed according to the actual initial distribution $\mu_{x,\xi}$ instead of some $\nu$. We discuss how to construct estimators for (8) in Algorithm 5 (Appendix D) and prove unbiasedness in Proposition E.3 (Appendix E). A special case of Equation (8) appeared in (Chen et al., 2022), where they consider deterministic model design for MDPs, that does not take into account contextual uncertainty or the possibility of multiple followers.

**C.2. Accelerated HPGD with Full Lower-Level Access**

Previously, we assumed that the leader does not know the solver used in the lower level and queries trajectories from an oracle. In certain situations, the leader can additionally influence how the followers solve the CMDP. Such settings appear in previous works (Chen et al., 2022), and in applications, such as dynamic mechanism design (Curry et al., 2024). In this Section, we argue how the additional assumption can be used to reduce the number of iterations the follower runs for the lower level.

Assume that the lower level is solved using Q-learning. The follower needs to run $T = \tilde{\mathcal{O}}(\delta^{-2})$ iterations to ensure that $\mathbb{E}\|\pi_{x,\xi}^T - \pi_{x,\xi}^*\|_\infty^2 \leq \delta^2$, where $\pi_{x,\xi}^t$ denotes the learned policy after $t$ Q-learning iterations. To reduce this number, we propose a randomized early stopping scheme over the lower-level iterations. Without loss of generality, consider a subsequence $t_k := 2^k$ such that $t_K := T$. Let $\frac{d}{dx}F_T$ denote the hypergradient estimator, based on the $T$-th policy iterate $\pi_{x,\xi}^T$. It holds that:

$$\frac{d}{dx}F_T = \frac{d}{dx}F_{t_K} = \frac{d}{dx}F_{t_1} + \sum_{k=1}^{K-1}\left(\frac{d}{dx}F_{t_{k+1}} - \frac{d}{dx}F_{t_k}\right) = \frac{d}{dx}F_{t_1} + \mathbb{E}_{\hat{k}\sim p_k}\left[\frac{\frac{d}{dx}F_{t_{\hat{k}+1}} - \frac{d}{dx}F_{t_{\hat{k}}}}{p_{\hat{k}}}\right],$$

where $p_k$ denotes a truncated geometric distribution, such that $p_{\hat{k}} \propto 2^{-\hat{k}}$. The above shows that $\frac{d}{dx}F_{t_1} + p_{\hat{k}}^{-1}\left[\frac{d}{dx}F_{t_{\hat{k}+1}} - \frac{d}{dx}F_{t_{\hat{k}}}\right]$ with $\hat{k} \sim p_k$ is an unbiased estimator of $\frac{d}{dx}F_T$. In this way, the follower does not need to run $2^K$ Q-learning iterations but in expectation only $\sum_{k=1}^{K-1} p_k t_k = \mathcal{O}(K)$. This implies that if the leader can direct how the follower learns, we can generate a hypergradient estimator with the same bias as $\frac{d}{dx}F_T$ but a much smaller lower-level iteration complexity of $\mathcal{O}(K)$ instead of $2^K$. We defer the proof of this observation to Proposition E.8 in Appendix E.

This idea has been studied for contextual bilevel optimization under the name randomly truncated multilevel Monte-Carlo (Hu et al., 2024). Note, the reduction in sample complexity generally comes at the expense of an increased variance of the hypergradient estimator. In (Hu et al., 2024), this increase is logarithmic as the lower-level problem is a static optimization problem and the data generated to estimate the hypergradient is independent from the lower-level decision. This structure is crucial for controlling the increased variance of the hypergradient estimator. In our problem, rollouts generated from $\pi_{x,\xi}^t$ are used to estimate the hypergradient. These trajectory samples thus depend on the lower-level decision and one may not be able to achieve the same variance as in (Hu et al., 2024). Nevertheless, the method allows us to significantly reduce the iterations that the follower runs per upper-level update at the potential expense of an increased variance. We leave the analysis of the overall variance and stationary convergence for future work.

## D. Algorithms

Here we give the pseudocode to certain algorithms/routines/procedures mentioned in the main text.

---

**Algorithm 2** `GradEst`

---

**Input:** $\xi, x$ state $s$, action $a$, trajectory oracle $o$

$T_Q, T_V \sim \text{Geo}(1 - \gamma), T'_Q, T'_V \sim \text{Geo}(1 - \gamma^{0.5})$

$\tau_Q \leftarrow \texttt{SampleTrajectory}(o, \texttt{start} = (s, a), \texttt{length} = T_Q + T'_Q + 1)$

$\tau_V \leftarrow \texttt{SampleTrajectory}(o, \texttt{start} = s, \texttt{length} = T_V + T'_V + 1)$

$\widehat{\frac{d}{dx}Q}(s,a) \leftarrow \sum_{t=0}^{T_Q} \frac{d}{dx} r(s_t^{\tau_Q}, a_t^{\tau_Q}) +$

$\frac{\gamma}{1-\gamma} \frac{d}{dx} \log P(s_{T_Q+1}^{\tau_Q}; s_{T_Q}^{\tau_Q}, a_{T_Q}^{\tau_Q}) \sum_{t=T_Q+1}^{T_Q+T'_Q+1} \gamma^{(t-T_Q-1)/2} \left( r(s_t^{\tau_Q}, a_t^{\tau_Q}) + \lambda H(\pi(\cdot; s_t)) \right)$

$\widehat{\partial_x V}(s) \leftarrow \sum_{t=0}^{T_V} \partial_x r(s_t^{\tau_V}, a_t^{\tau_V}) +$

$\frac{\gamma}{1-\gamma} \partial_x \log P(s_{T_V+1}^{\tau_V}; s_{T_V}^{\tau_V}, a_{T_V}^{\tau_V}) \sum_{t=T_V+1}^{T_V+T'_V+1} \gamma^{(t-T_V-1)/2} \left( r(s_t^{\tau_V}, a_t^{\tau_V}) + \lambda H(\pi(\cdot; s_t)) \right)$

**Output:** $\widehat{\partial_x A(s,a)} \leftarrow \widehat{\partial_x Q}(s,a) - \widehat{\partial_x V}(s)$

---

**Algorithm 3** Soft Value Iteration

---

1: **Input:** Number of iterations $T$

2: **Result:** Approximation $V_\lambda \approx V_\lambda^*$, policy $\pi_\lambda \approx \pi_\lambda^*$

3: Initialize $V_\lambda = 0$

4: **for** $t = 0$ to $T$ **do**

5:    **for** $s \in \mathcal{S}$ **do**

6:       **for** $a \in \mathcal{A}$ **do**

7:          $Q_\lambda(s,a) = r(s,a) + \gamma \mathbb{E}_{s'|s,a} \left[ V_\lambda(s') \right]$

8:       **end for**

9:       $V_{\text{new},\lambda}(s) = \lambda \log \left( \sum_{a \in \mathcal{A}} \exp \left( \frac{Q_\lambda(s,a)}{\lambda} \right) \right)$

10:    **end for**

11:    set $V_\lambda := V_{\text{new},\lambda}$

12: **end for**

13: $\pi_\lambda^o \leftarrow \frac{\exp(Q_\lambda(a|s)/\lambda)}{\sum_a \exp(Q_\lambda(a|s)/\lambda)}$

14: **return** $V_\lambda$ and $\pi_\lambda^o$

---

**Algorithm 4** Soft Q-learning

---

1: **Input:** Number of iterations $T$, Behavioural Policy $\pi_B$, Stepsizes $\{\alpha_t\}_{t \geq 0}$

2: **Result:** Approximation $Q_\lambda \approx Q_\lambda^*$, policy $\pi_\lambda \approx \pi_\lambda^*$

3: Initialize $Q_\lambda = 0$

4: Initialise $s_0$

5: **for** $t = 0$ to $T$ **do**

6:    Sample $a \sim \pi_B(\cdot; s_t)$

7:    Observe next reward $r(s_t, a)$ and state $s_{t+1} \sim P(\cdot|s_t, a)$

8:    $Q_\lambda(s_t, a) = Q_\lambda(s_t, a) + \alpha_t \left( r(s_t, a) + \gamma \lambda \log \left( \sum_{a' \in \mathcal{A}} \exp \left( \frac{Q_\lambda(s_{t+1}, a')}{\lambda} \right) \right) \right)$

9: **end for**

10: $\pi_\lambda^o(a; s) \leftarrow \frac{\exp(Q_\lambda(a|s)/\lambda)}{\sum_{a'} \exp(Q_\lambda(a'|s)/\lambda)}$

11: **return** $Q_\lambda$ and $\pi_\lambda^o$

---

---

**Algorithm 5** `DecomposableGradientEstimator`

---

**Input:** $\xi, x$, initial distribution $\mu_{x,\xi}$, oracle $o$

$T, \sim \mathrm{Geo}(1 - \gamma), T' \sim \mathrm{Geo}(1 - \gamma^{0.5})$

$(s_0, a_0, \ldots, s_{T+T'}, a_{T+T'}) \leftarrow \mathtt{SampleTrajectory}(o, \mathtt{start} = \mu_{x,\xi}, \mathtt{length} = T + T')$

$\widehat{A_{\lambda,x,\xi}^{\pi_{x,\xi}^o}}(s_T, a_T) \leftarrow \mathtt{GradientEstimator}(\xi, x, s_T, a_T, o)$

$\widehat{\frac{dF}{dx}} = \left( \sum_{t=0}^{T} \frac{d}{dx} \overline{r}(s_t, a_t) \right) + \frac{1}{\lambda(1-\gamma)} \partial_x \widehat{A_{\lambda,x,\xi}^{\pi_{x,\xi}^o}}(s_T, a_T) \sum_{t'=T}^{T+T'} \gamma^{(t-T)/2} \overline{r}(s_{t'}, a_{t'})$

$\qquad + \frac{1}{1-\gamma} \partial_x \log P(s_T, a_{T-1}, s_{T-1}) \sum_{t'=T}^{T+T'} \gamma^{(t'-T)/2} \overline{r}(s_{t'}, a_{t'})$

**Output:** $\widehat{\frac{dF}{dx}}$

---

---

**Algorithm 6** Vanilla Policy Gradient Algorithm

---

**Data:** Initial parameter $\theta_0$, initial state $s$

**Result:** Approximate policy $\pi_{\theta_L}$

**for** $l = 0$ to $L$ **do**

$\qquad$ Sample $T \sim \mathrm{Geo}(1 - \gamma)$

$\qquad$ Sample trajectory $(s_0, a_0, s_1, \ldots, a_{T-1}, s_T, r_T, a_T)$ using policy $\pi_{\theta_l}$

$\qquad$ Sample $T' \sim \mathrm{Geo}(1 - \gamma^2)$

$\qquad$ Set $\tilde{s}_0 = s_{T'}$ and $\tilde{a}_0 = a_T$

$\qquad$ Sample trajectory $(\tilde{s}_0, \tilde{a}_0, \tilde{s}_1, \ldots, \tilde{a}_{T'-1}, \tilde{s}_{T'}, \tilde{r}_{T'}, \tilde{a}_{T'})$ using policy $\pi_{\theta_l}$

$\qquad$ Determine step-size $\alpha$.

$\qquad \widehat{\nabla \tilde{J}_s}(\theta_l) = \frac{1}{1-\gamma} \nabla \log \pi_{\theta_l}(a_T | s_T) \sum_{t'=0}^{T'-1} \gamma^{t'/2} \tilde{r}_{t'+1}$

$\qquad \theta_{l+1} = \theta_l - \alpha \widehat{\nabla \tilde{J}_s}(\theta_l)$

**end for**

---

# E. Proofs

In this section, we provide the proofs for the presented Theorems and Propositions. We provide the proof of Theorem 2.2, deriving the hypergradient of function $F(x)$; the proof of Theorem 2.3 which derives the derivative of the action-value function with respect to $x$; the proof of our main result, Theorem 2.5, which shows convergence of HPGD to a stationary point of $F(x)$.

For the Propositions we show how to estimate the upper-level gradient if $f$ is decomposable in the proof of Proposition C.1; Proposition E.1 which shows how to compute the gradient of the optimal policy with respect to $x$; the proof of Proposition E.2, which shows we can achieve unbiased estimates of the advantage gradient; equivalently the proof of Proposition E.3, which shows the same for the special case when $f$ decomposes.

We state and proof Propositions E.4 to E.7 which show convergence in $L_2$ to the optimal policy of value iteration, Q-learning, Vanilla Policy Gradient and Natural Policy Gradient respectively.

Last, we state and proof Proposition E.8, which proves the reduced iteration complexity claimed in Appendix C.2.

***Theorem 2.2.*** Note that from Proposition 2.3 it follows that

$$\left\| \frac{\partial \pi_{x,\xi}^*}{\partial x} \right\|_\infty \leq \frac{2}{\lambda} \left\| \partial_x Q_{\lambda,x,\xi}^{\pi_{x,\xi}^*}(s, a) \right\|_\infty$$

$$\leq \frac{2}{\lambda} \frac{K_2}{1 - \gamma} \frac{K_1 \overline{R} K_1}{(1 - \gamma)^2}$$

Therefore we can apply DCT to get

$$\partial_x \mathbb{E}\left[f(x, \pi_{x,\xi}^*, \xi)\right] = \mathbb{E}\left[\partial_x f(x, \pi_{x,\xi}^*, \xi)\right]$$

$$= \mathbb{E}\left[\frac{\partial_1 f(x, \pi_{x,\xi}^*, \xi)}{\partial x} + \frac{\partial_2 f(x, \pi_{x,\xi}^*, \xi)}{\partial \pi_{x,\xi}^*}\frac{\partial \pi_{x,\xi}^*}{\partial x}\right]$$

$$= \mathbb{E}\left[\frac{\partial_1 f(x, \pi_{x,\xi}^*, \xi)}{\partial x} + \sum_{s,a}\frac{\partial_2 f(x, \pi_{x,\xi}^*, \xi)}{\partial \pi_{x,\xi}^*(a;s)}\frac{\partial \pi_{x,\xi}^*(a;s)}{\partial x}\right]$$

$$= \mathbb{E}\left[\frac{\partial_1 f(x, \pi_{x,\xi}^*, \xi)}{\partial x} + \sum_{s,a}\frac{\partial_2 f(x, \pi_{x,\xi}^*, \xi)}{\partial \pi_{x,\xi}^*(a;s)}\frac{1}{\lambda}\pi_{x,\xi}^*(a;s)\partial_x A_{\lambda,x,\xi}^{\pi_{x,\xi}^*}(s,a)\right] \quad (9)$$

$$= \mathbb{E}\left[\frac{\partial_1 f(x, \pi_{x,\xi}^*, \xi)}{\partial x} + \mathbb{E}_{s\sim\nu, a\sim\pi_{x,\xi}^*}\left[\frac{1}{\lambda\nu(s)}\frac{\partial_2 f(x, \pi_{x,\xi}^*, \xi)}{\partial \pi_{x,\xi}^*(a;s)}\partial_x A_{\lambda,x,\xi}^{\pi_{x,\xi}^*}(s,a)\right]\right]$$

where we use Proposition E.1 for Equation (9). Further, we note that $\frac{\partial_2 f(x, \pi_{x,\xi}^*, \xi)}{\partial \pi_{x,\xi}^*} \in \mathrm{Mat}_{1,|S|\times|A|}(\mathbb{R})$ and $\frac{\partial \pi_{x,\xi}^*}{\partial x} \in \mathrm{Mat}_{|S|\times|A|,d}(\mathbb{R})$, such that we just explicitly write out the matrix multiplication for the second equality. $\qquad\square$

***Theorem 2.3***. We show the equivalent formulation

$$\frac{dQ_{\lambda,x,\xi}^\pi(s,a)}{dx} = \sum_{t=0}^\infty\sum_{s',a'}\gamma^t p_{x,\xi}(s,a \to s',a';t,\pi)\left(\frac{dr_{x,\xi}(s',a')}{dx} + \gamma\sum_{s''}\frac{dP_{x,\xi}(s'';s',a')}{dx}V_{\lambda,x,\xi}^\pi(s'')\right)$$

where $p_{x,\xi}(s,a \to s',a';t,\pi)$ is the probably that starting from $s,a$ the Markov Chain induced by $\pi$ reaches $s',a'$ after $t$ steps.

The proof follows the proof of the standard policy gradient theorem. Note that we drop here the dependence on $x$ and $\xi$ to simplify the notation. Assuming the derivative exists at each state action pair, we will show by induction that for all $n \in \mathbb{N}$ it holds that

$$\frac{dQ_\lambda^\pi(s,a)}{dx} = \sum_{t=0}^n\sum_{s',a'}\gamma^t p(s,a \to s',a';t,\pi)\left(\frac{dr(s',a')}{dx} + \gamma\sum_{s''}\frac{dP(s'';s',a')}{dx}V_\lambda^\pi(s'')\right)$$
$$+ \gamma^{n+1}\sum_{\tilde{s},\tilde{a}} p(s,a \to \tilde{s},\tilde{a};n+1,\pi)\frac{dQ_\lambda^\pi(\tilde{s},\tilde{a})}{dx}$$

The claim then follows by considering the limit as $n \to \infty$.

**Base case** $(n = 0)$    It is easy to check that

$$\frac{dQ_\lambda^\pi(s,a)}{dx} = \frac{d}{dx}\left(r(s,a) + \gamma \sum_{s'} P(s';s,a)V_\lambda(s')\right)$$

$$= \frac{d}{dx}r(s,a) + \gamma \sum_{s'}(\frac{d}{dx}P(s';s,a)V_\lambda(s') + P(s';s,a)\frac{d}{dx}V_\lambda(s'))$$

$$= \frac{d}{dx}r(s,a) + \gamma \sum_{s'}\left(\frac{d}{dx}P(s';s,a)V_\lambda(s') + P(s';s,a)\frac{d}{dx}V_\lambda(s')\right)$$

$$= \frac{d}{dx}r(s,a) + \gamma \sum_{s'}\left(\frac{d}{dx}P(s';s,a)V_\lambda(s') + P(s';s,a)\sum_{a'}\pi(a';s')\frac{d}{dx}Q_\lambda(s',a')\right)$$

$$= \sum_{t=0}^{0}\sum_{s',a'}\gamma^t p(s,a \to s',a';t,\pi)\left(\frac{dr(s',a')}{dx} + \gamma \sum_{s''}\frac{dP(s'';s',a')}{dx}V_\lambda^\pi(s'')\right)$$

$$+ \gamma^1 \sum_{\tilde{s},\tilde{a}} p(s,a \to \tilde{s},\tilde{a};n+1,\pi)\frac{dQ_\lambda^\pi(\tilde{s},\tilde{a})}{dx}$$

**Induction step** $(n \implies n+1)$

$$\frac{dQ_\lambda^\pi(s,a)}{dx} = \sum_{t=0}^{n}\sum_{s',a'}\gamma^t p(s,a \to s',a';t,\pi)\left(\frac{dr(s',a')}{dx} + \gamma \sum_{s''}\frac{dP(s'';s',a')}{dx}V_\lambda^\pi(s'')\right)$$

$$+ \gamma^{n+1} \sum_{\tilde{s},\tilde{a}} p(s,a \to \tilde{s},\tilde{a};n+1,\pi)\frac{dQ_\lambda^\pi(\tilde{s},\tilde{a})}{dx}$$

$$= \sum_{t=0}^{n}\sum_{s',a'}\gamma^t p(s,a \to s',a';t,\pi)\left(\frac{dr(s',a')}{dx} + \gamma \sum_{s''}\frac{dP(s'';s',a')}{dx}V_\lambda^\pi(s'')\right)$$

$$+ \gamma^{n+1} \sum_{\tilde{s},\tilde{a}} p(s,a \to \tilde{s},\tilde{a};n+1,\pi)\frac{d}{dx}\left(r(\tilde{s},\tilde{a}) + \gamma \sum_{\tilde{s}'}P(\tilde{s}';\tilde{s},\tilde{a})V_\lambda(\tilde{s}')\right)$$

$$= \sum_{t=0}^{n}\sum_{s',a'}\gamma^t p(s,a \to s',a';t,\pi)\left(\frac{dr(s',a')}{dx} + \gamma \sum_{s''}\frac{dP(s'';s',a')}{dx}V_\lambda^\pi(s'')\right)$$

$$+ \gamma^{n+1} \sum_{\tilde{s},\tilde{a}} p(s,a \to \tilde{s},\tilde{a};n+1,\pi)\left(\frac{d}{dx}r(\tilde{s},\tilde{a}) + \gamma \sum_{\tilde{s}'}\frac{d}{dx}P(\tilde{s}';\tilde{s},\tilde{a})V_\lambda(\tilde{s}')\right.$$

$$\left. + P(\tilde{s}';\tilde{s},\tilde{a})\sum_{\tilde{a}'}\pi(\tilde{a}';\tilde{s}')\frac{d}{dx}Q_\lambda(\tilde{s}',\tilde{a}')\right)$$

$$= \sum_{t=0}^{n+1}\sum_{s',a'}\gamma^t p(s,a \to s',a';t,\pi)\left(\frac{dr(s',a')}{dx} + \gamma \sum_{s''}\frac{dP(s'';s',a')}{dx}V_\lambda^\pi(s'')\right)$$

$$+ \gamma^{n+2} \sum_{\tilde{s},\tilde{a}} p(s,a \to \tilde{s},\tilde{a};n+2,\pi)\frac{dQ_\lambda^\pi(\tilde{s},\tilde{a})}{dx}$$

$\square$

**Theorem 2.5**. By smoothness of $f$, we can use the following bound from (Hu et al., 2024):

$$\mathbb{E}\left[\left\|\frac{dF(\hat{x}_T)}{dx}\right\|_\infty^2\right] \leq \underbrace{\frac{2(F(x_1) - \min_x F(x))}{\alpha T}}_{(1)} + \frac{2}{T}\sum_{t=1}^{T}\left(L\underbrace{\left\|\mathbb{E}\left[\frac{dF(x_t)}{dx} - \frac{\widehat{dF(x_t)}}{dx}\right]\right\|_\infty}_{(2)}\right.$$

$$\left. + 2S_f\alpha\underbrace{\mathbb{E}\left[\left\|\frac{dF(x_t)}{dx} - \frac{\widehat{dF(x_t)}}{dx}\right\|_\infty^2\right]}_{(3)}\right). \tag{10}$$

The error term naturally decomposes into an initial error, which decreases with $T$, a bias term, and a variance term, which decreases with the stepsize $\alpha$.

For **(1)** we do not need to simplify any further.

Next up, we decompose the bias term **(2)**

$$\left\|\mathbb{E}\left[\frac{dF(x_t)}{dx} - \frac{\widehat{dF(x_t)}}{dx}\right]\right\|_\infty$$

$$\leq \left\|\mathbb{E}_{x_t}\left[\frac{dF(x_t)}{dx} - \mathbb{E}_{\xi,o}\left[\frac{\partial_1 f(x_t, \pi^o_{x_t,\xi}, \xi)}{\partial x} + \mathbb{E}^{a\sim\pi^o_{x_t,\xi}}_\nu\left[\frac{1}{\lambda\nu(s)}\frac{\partial_2 f(x_t, \pi^o_{x_t,\xi}, \xi))}{\partial\pi(s,a)}\mathbb{E}\left[\frac{\widehat{d}}{dx}A^{\pi^o_{x_t,\xi}}_{\lambda,x,\xi}(s,a)\right]\right]\right]\right]\right\|_\infty$$

$$\leq \underbrace{\left\|\mathbb{E}_{x_t,o,\xi}\left[\frac{\partial_1 f(x_t, \pi^*_{x_t,\xi}, \xi)}{\partial x} - \frac{\partial_1 f(x_t, \pi^o_{x_t,\xi}, \xi)}{\partial x}\right]\right\|_\infty}_{(A)}$$

$$+ \underbrace{\left\|\mathbb{E}^{a\sim\pi^*(x,\xi)}_{x_t,o,\xi,\nu}\left[\frac{1}{\lambda\nu(s)}\left(\frac{\partial_2 f(x, \pi^*_{x,\xi}, \xi)}{\partial\pi^*_{x,\xi}(a;s)}\partial_x A^{\pi^*_{x,\xi}}_{\lambda,x,\xi}(s,a) - \frac{\partial_2 f(x, \pi^o_{x_t,\xi}, \xi)}{\partial\pi^o_{x_t,\xi}(a;s)}\partial_x A^{\pi^o_{x_t,\xi}}_{\lambda,x,\xi}(s,a)\right)\right]\right\|_\infty}_{(B)}$$

Where we use that $\frac{\widehat{d}}{dx}A^{\pi^o_{x_t,\xi}}_{\lambda,x,\xi}$ is an unbiased estimator of $\frac{d}{dx}A^{\pi^o_{x_t,\xi}}_{\lambda,x,\xi}$ as shown in Proposition E.2.

**(A)** is relatively easy to bound. Indeed by the smoothness of $f$ ( Assumption 2.1) it immediately follows that

$$(A) \leq \mathbb{E}_{x_t,o,\xi}\left[S_f\left\|\pi^*_{x_t,\xi} - \pi^o_{x_t,\xi}\right\|_\infty\right] \leq S_f\delta$$

To bound **(B)** we further decompose it

$$(B) = \left\|\mathbb{E}^{\xi,\nu}_{x_t,o}\left[\frac{1}{\lambda\nu(s)}\sum_a\left(\pi^*_{x,\xi}(a;s)\frac{\partial_2 f(x, \pi^*_{x,\xi}, \xi)}{\partial\pi^*_{x,\xi}(a;s)}\partial_x A^{\pi^*_{x,\xi}}_{\lambda,x,\xi}(s,a) - \pi^o_{x,\xi}(a;s)\frac{\partial_2 f(x, \pi^o_{x_t,\xi}, \xi)}{\partial\pi^o_{x_t,\xi}(a;s)}\partial_x A^{\pi^o_{x_t,\xi}}_{\lambda,x,\xi}(s,a)\right)\right]\right\|_\infty$$

$$\leq \underbrace{\mathbb{E}^{\xi,\nu}_{x_t,o}\left[\frac{1}{\lambda\nu(s)}\sum_a\left\|\pi^*_{x,\xi}(a;s) - \pi^o_{x,\xi}(a;s)\right\|_\infty\left\|\frac{\partial_2 f(x, \pi^*_{x,\xi}, \xi)}{\partial\pi^*_{x,\xi}(a;s)}\partial_x A^{\pi^*_{x,\xi}}_{\lambda,x,\xi}(s,a)\right\|_\infty\right]}_{(a)}$$

$$+ \underbrace{\mathbb{E}^{\xi,\nu}_{x_t,o}\left[\frac{1}{\lambda\nu(s)}\sum_a\left\|\pi^o_{x,\xi}(a;s)\right\|_\infty\left\|\frac{\partial_2 f(x, \pi^*_{x_t,\xi}, \xi)}{\partial\pi^*_{x_t,\xi}(a;s)}\partial_x A^{\pi^*_{x_t,\xi}}_{\lambda,x,\xi}(s,a) - \frac{\partial_2 f(x, \pi^o_{x_t,\xi}, \xi)}{\partial\pi^o_{x_t,\xi}(a;s)}\partial_x A^{\pi^o_{x_t,\xi}}_{\lambda,x,\xi}(s,a)\right\|_\infty\right]}_{(b)}$$

**(a)** is relatively easy to bound. We have

$$\textbf{(a)} \leq \mathbb{E}_{x_t}^{\xi,\nu}\left[\frac{1}{\lambda\nu(s)}|\mathcal{A}|\delta\left\|\frac{\partial_2 f(x,\pi_{x,\xi}^*,\xi)}{\partial\pi_{x,\xi}^*(a;s)}\partial_x A_{\lambda,x,\xi}^{\pi_{x,\xi}^*}(s,a)\right\|_\infty\right]$$

$$\leq \mathbb{E}_{x_t}^{\xi,\nu}\left[\frac{1}{\lambda\nu(s)}|\mathcal{A}|\delta L_f\left\|\partial_x A_{\lambda,x,\xi}^{\pi_{x,\xi}^*}(s,a)\right\|_\infty\right]$$

$$\leq \mathbb{E}_{x_t}^{\xi,\nu}\left[\frac{1}{\lambda\nu(s)}|\mathcal{A}|\delta L_f\left\|\partial_x A_{\lambda,x,\xi}^{\pi_{x,\xi}^*}(s,a)\right\|_\infty\right]$$

$$\leq \mathbb{E}_{x_t}^{\xi,\nu}\left[\frac{1}{\lambda\nu(s)}|\mathcal{A}|\delta L_f 2\left\|\partial_x Q_{\lambda,x,\xi}^{\pi_{x,\xi}^*}(s,a)\right\|_\infty\right]$$

$$= \mathbb{E}_{x_t}^{\xi,\nu}\left[\frac{1}{\lambda\nu(s)}|\mathcal{A}|\delta L_f 2\left\|\mathbb{E}_{s,a}^{\pi_{x,\xi}^*}\left[\sum_{t=0}^{\infty}\gamma^t\frac{dr_{x,\xi}(s_t,a_t)}{dx}+\gamma^{t+1}\frac{d\log P_{x,\xi}(s_{t+1};s_t,a_t)}{dx}V_{\lambda,x,\xi}^{\pi}(s_{t+1})\right]\right\|_\infty\right]$$

where we use the assumption on the oracle and that $f$ is Lipschitz. Note that it holds that

$$\left\|V_{\lambda,x,\xi}^{\pi}(s)\right\|_\infty \leq \frac{(\overline{R}+\lambda\log|\mathcal{A}|)}{1-\gamma}$$

And thus

$$\left\|\partial_x Q_{\lambda,x,\xi}^{\pi_{x,\xi}^*}(s,a)\right\|_\infty \leq \left(\frac{K_2}{1-\gamma}+\frac{K_1(\overline{R}+\lambda\log|\mathcal{A}|)}{(1-\gamma)^2}\right)$$

Letting $m := \min_s \nu(s)$ we thus have

$$\textbf{(a)} \leq \frac{1}{\lambda m}|\mathcal{A}|\delta L_f 2\left(\frac{K_2}{1-\gamma}+\frac{K_1(\overline{R}+\lambda\log|\mathcal{A}|)}{(1-\gamma)^2}\right)$$

For **(b)** we further simplify

$$\textbf{(b)} \leq \frac{1}{\lambda m}\mathbb{E}_{x_t,o}^{\xi}\left[\left\|\frac{\partial_2 f(x,\pi_{x_t,\xi}^*,\xi)}{\partial\pi_{x_t,\xi}^*(a;s)}\partial_x A_{\lambda,x,\xi}^{\pi_{x_t,\xi}^*}(s,a)-\frac{\partial_2 f(x,\pi_{x_t,\xi}^o,\xi)}{\partial\pi_{x_t,\xi}^o(a;s)}\partial_x A_{\lambda,x,\xi}^{\pi_{x_t,\xi}^o}(s,a)\right\|_\infty\right]$$

$$\leq \underbrace{\frac{1}{\lambda m}\mathbb{E}_{x_t,o}^{\xi}\left[\left\|\frac{\partial_2 f(x,\pi_{x_t,\xi}^*,\xi)}{\partial\pi_{x_t,\xi}^*(a;s)}-\frac{\partial_2 f(x,\pi_{x_t,\xi}^o,\xi)}{\partial\pi_{x_t,\xi}^o(a;s)}\right\|_\infty\left\|\partial_x A_{\lambda,x,\xi}^{\pi_{x_t,\xi}^*}(s,a)\right\|_\infty\right]}_{\textbf{(i)}}$$

$$+ \underbrace{\frac{1}{\lambda m}\mathbb{E}_{x_t,o}^{\xi}\left[\left\|\frac{\partial_2 f(x,\pi_{x_t,\xi}^o,\xi)}{\partial\pi_{x_t,\xi}^o(a;s)}\right\|_\infty\left\|\partial_x A_{\lambda,x,\xi}^{\pi_{x_t,\xi}^*}(s,a)-\partial_x A_{\lambda,x,\xi}^{\pi_{x_t,\xi}^o}(s,a)\right\|_\infty\right]}_{\textbf{(ii)}}$$

Similar to **(a)**, we can bound **(i)** using Assumption 2.1.

$$\textbf{(i)} \leq \frac{S_f}{\lambda m}\delta 2\left(\frac{K_2}{1-\gamma}+\frac{K_1(\overline{R}+\lambda\log|\mathcal{A}|)}{(1-\gamma)^2}\right)$$

Bounding **(ii)** is the tricky part of this proof. We first need to show two intermediate results. First, we bound the difference in entropy between two policies. For the entropy we denote by $l_1 := \min_{s,a,x,\xi}\pi_{x,\xi}^*(a;s)$ the minimum probability of playing an action in any state under the optimal policy. Note that

$$l_1 \geq \frac{\exp(\frac{-\overline{R}}{\lambda(1-\gamma)})}{|\mathcal{A}|\exp(\frac{\overline{R}}{\lambda(1-\gamma)})} > 0$$

We assume now that $\delta$ is sufficiently small, i.e. $\delta \le l_1/2$, such that $l_1/2 \le \min_{s,a,x,\xi} \pi^o_{x,\xi}(a;s)$

Note that $\log$ is Lipschitz with parameter $\frac{1}{a}$ on $[a, \infty)$. Hence we have

$$
\left\| H(\pi^*_{x,\xi}|s) - H(\pi^o_{x,\xi}|s) \right\|_\infty
$$
$$
= \left\| \sum_a \pi^*_{x,\xi}(a;s) \log \pi^*_{x,\xi}(a;s) - \sum_a \pi^o_{x,\xi}(a;s) \log \pi^o_{x,\xi}(a;s) \right\|_\infty
$$
$$
\le \left( \sum_a \left\| \pi^*_{x,\xi} - \pi^o_{x,\xi} \right\|_\infty \left\| \log \pi^*_{x,\xi} \right\|_\infty \right) + \left\| \log \pi^*_{x,\xi} - \log \pi^o_{x,\xi} \right\|_\infty
$$
$$
\le |\mathcal{A}||\log l_1|\delta + \frac{2}{l_1}\delta
$$

Then we use this to bound the difference in the value functions.

$$
\left\| V^{\pi^*_{x,\xi}}_\lambda(s) - V^{\pi^*_{x,\xi}}_\lambda \right\|_\infty \le \left\| \sum_a \left( \pi^*_{x,\xi}(a;s) Q^{\pi^*_{x,\xi}}_\lambda(s,a) - \pi^o_{x,\xi}(a;s) Q^{\pi^o_{x,\xi}}_\lambda(s,a) \right) \right\|_\infty + \lambda\delta \left( |\mathcal{A}||\log l_1| + \frac{2}{l_1} \right)
$$
$$
\le \left\| \sum_a \left( \pi^*_{x,\xi}(a;s) \right) \right\|_\infty \left\| Q^{\pi^*_{x,\xi}}_\lambda(s,a) - Q^{\pi^o_{x,\xi}}_\lambda(s,a) \right\|_\infty
$$
$$
+ \sum_a \left\| \pi^*_{x,\xi}(a;s) - \pi^o_{x,\xi}(a;s) \right\|_\infty \left\| Q^{\pi^o_{x,\xi}}_\lambda(s,a) \right\|_\infty + \lambda\delta \left( |\mathcal{A}||\log l_1| + \frac{2}{l_1} \right)
$$
$$
\le \lambda\delta \left( |\mathcal{A}||\log l_1| + \frac{2}{l_1} \right) + \delta|\mathcal{A}|\frac{\overline{R}}{1-\gamma} + \gamma \left\| V^{\pi^*_{x,\xi}}_\lambda(s) - V^{\pi^o_{x,\xi}}_\lambda(s) \right\|_\infty
$$
$$
\le \frac{\lambda\delta \left( |\mathcal{A}||\log l_1| + \frac{2}{l_1} \right)}{1-\gamma} + \frac{\delta|\mathcal{A}|\overline{R}}{(1-\gamma)^2}
$$

Now we employ a similar technique again to bound (ii) using the above results

$$
\frac{1}{\lambda m} \mathbb{E}^\xi_{x_t,o} \left[ \left\| \frac{\partial_2 f(x, \pi^o_{x_t,\xi}, \xi)}{\partial \pi^o_{x_t,\xi}(a;s)} \right\|_\infty \left\| \partial_x A^{\pi^*_{x_t,\xi}}_{\lambda,x,\xi}(s,a) - \partial_x A^{\pi^o_{x_t,\xi}}_{\lambda,x,\xi}(s,a) \right\|_\infty \right]
$$
$$
\le \frac{L_f}{\lambda m} \mathbb{E}^\xi_{x_t,o} \left[ \left\| \partial_x A^{\pi^*_{x_t,\xi}}_{\lambda,x,\xi}(s,a) - \partial_x A^{\pi^o_{x_t,\xi}}_{\lambda,x,\xi}(s,a) \right\|_\infty \right]
$$
$$
\le \frac{L_f}{\lambda m} 2\mathbb{E}^\xi_{x_t,o} \left[ \left\| \partial_x Q^{\pi^*_{x_t,\xi}}_{\lambda,x,\xi}(s,a) - \partial_x Q^{\pi^o_{x_t,\xi}}_{\lambda,x,\xi}(s,a) \right\|_\infty \right]
$$

We bound the difference in derivatives

$$\left\| \partial_x Q_{\lambda,x,\xi}^{\pi_{x_t,\xi}^*}(s,a) - \partial_x Q_{\lambda,x,\xi}^{\pi_{x_t,\xi}^o}(s,a) \right\|_\infty$$

$$= \left\| \sum_{t=0}^\infty \sum_{s',a'} \gamma^t p_{x,\xi}(s,a \to s',a';t,\pi_{x_t,\xi}^*) \left( \frac{dr_{x,\xi}(s',a')}{dx} + \gamma \sum_{s''} \frac{dP_{x,\xi}(s'';s',a')}{dx} V_{\lambda,x,\xi}^{\pi_{x_t,\xi}^*}(s'') \right) \right.$$

$$\left. - \sum_{t=0}^\infty \sum_{s',a'} \gamma^t p_{x,\xi}(s,a \to s',a';t,\pi_{x_t,\xi}^o) \left( \frac{dr_{x,\xi}(s',a')}{dx} + \gamma \sum_{s''} \frac{dP_{x,\xi}(s'';s',a')}{dx} V_{\lambda,x,\xi}^{\pi_{x_t,\xi}^o}(s'') \right) \right\|_\infty$$

$$= \left\| \frac{dr_{x,\xi}(s,a)}{dx} + \gamma \sum_{s'} \frac{dP_{x,\xi}(s';s,a)}{dx} V_{\lambda,x,\xi}^{\pi_{x_t,\xi}^*}(s') \right.$$

$$+ \sum_{t=1}^\infty \sum_{s',a'} \gamma^t p_{x,\xi}(s,a \to s',a';t,\pi_{x_t,\xi}^*) \left( \frac{dr_{x,\xi}(s',a')}{dx} + \gamma \sum_{s''} \frac{dP_{x,\xi}(s'';s',a')}{dx} V_{\lambda,x,\xi}^{\pi_{x_t,\xi}^*}(s'') \right)$$

$$- \frac{dr_{x,\xi}(s,a)}{dx} - \gamma \sum_{s'} \frac{dP_{x,\xi}(s';s,a)}{dx} V_{\lambda,x,\xi}^{\pi_{x_t,\xi}^o}(s')$$

$$\left. - \sum_{t=1}^\infty \sum_{s',a'} \gamma^t p_{x,\xi}(s,a \to s',a';t,\pi_{x_t,\xi}^o) \left( \frac{dr_{x,\xi}(s',a')}{dx} + \gamma \sum_{s''} \frac{dP_{x,\xi}(s'';s',a')}{dx} V_{\lambda,x,\xi}^{\pi_{x_t,\xi}^o}(s'') \right) \right\|_\infty$$

$$\leq \gamma \sum_{s'} \left\| \frac{dP_{x,\xi}(s';s,a)}{dx} \right\|_\infty \left\| V_{\lambda,x,\xi}^{\pi_{x_t,\xi}^o}(s') - V_{\lambda,x,\xi}^{\pi_{x_t,\xi}^o}(s') \right\|_\infty$$

$$+ \gamma \left\| \sum_{s',a'} P(s';s,a)\pi_{x_t,\xi}^*(a',s') \sum_{t=0}^\infty \sum_{s',a'} \gamma^t p_{x,\xi}(s',a' \to s'',a'';t,\pi_{x_t,\xi}^*) \ldots \right.$$

$$\left( \frac{dr_{x,\xi}(s'',a'')}{dx} + \gamma \sum_{s'''} \frac{dP_{x,\xi}(s''';s'',a'')}{dx} V_{\lambda,x,\xi}^{\pi_{x_t,\xi}^*}(s'') \right)$$

$$- \sum_{s',a'} P(s';s,a)\pi_{x_t,\xi}^o(a',s') \sum_{t=0}^\infty \sum_{s',a'} \gamma^t p_{x,\xi}(s',a' \to s'',a'';t,\pi_{x_t,\xi}^o) \ldots$$

$$\left. \left( \frac{dr_{x,\xi}(s'',a'')}{dx} + \gamma \sum_{s'''} \frac{dP_{x,\xi}(s''';s'',a'')}{dx} V_{\lambda,x,\xi}^{\pi_{x_t,\xi}^o}(s'') \right) \right\|_\infty$$

$$\leq \gamma \sum_{s'} \left\| \frac{dP_{x,\xi}(s';s,a)}{dx} \right\|_\infty \left\| V_{\lambda,x,\xi}^{\pi_{x_t,\xi}^o}(s') - V_{\lambda,x,\xi}^{\pi_{x_t,\xi}^o}(s') \right\|_\infty$$

$$+ \gamma \left\| \sum_{s',a'} P(s';s,a)\pi_{x_t,\xi}^*(a',s')\partial_x Q_{\lambda,x,\xi}^{\pi_{x_t,\xi}^*}(s,a) \right.$$

$$\left. - \sum_{s',a'} P(s';s,a)\pi_{x_t,\xi}^o(a',s')\partial_x Q_{\lambda,x,\xi}^{\pi_{x_t,\xi}^o}(s,a) \right\|_\infty$$

$$\leq \gamma \sum_{s'} \left\| \frac{dP_{x,\xi}(s';s,a)}{dx} \right\|_\infty \left\| V_{\lambda,x,\xi}^{\pi_{x_t,\xi}^o}(s') - V_{\lambda,x,\xi}^{\pi_{x_t,\xi}^o}(s') \right\|_\infty$$

$$+ \gamma \left\| Q_{\lambda,x,\xi}^{\pi_{x_t,\xi}^*}(s',a') \right\|_\infty \left\| \sum_{s',a'} P(s';s,a) \right\|_\infty \left\| \pi_{x_t,\xi}^*(a',s') - \pi_{x_t,\xi}^o(a',s') \right\|_\infty$$

$$+ \gamma \sum_{s',a'} P(s';s,a)\pi_{x_t,\xi}^o(a',s') \left\| \partial_x Q_{\lambda,x,\xi}^{\pi_{x_t,\xi}^*}(s',a') - \partial_x Q_{\lambda,x,\xi}^{\pi_{x_t,\xi}^o}(s',a') \right\|_\infty$$

Where the dots indicate multiplication over the linebreak. Taking the expectation we thus get

$$\mathbb{E}_{x_t,o}^{\xi}\left[\left\|\partial_x Q_{\lambda,x,\xi}^{\pi_{x_t,\xi}^*}(s,a) - \partial_x Q_{\lambda,x,\xi}^{\pi_{x_t,\xi}^o}(s,a)\right\|_{\infty}\right]$$

$$\leq \gamma\left(|\mathcal{S}|K_1\left(\frac{2\lambda\log|\mathcal{A}|}{1-\gamma} + \frac{\delta|\mathcal{A}|\overline{R}}{(1-\gamma)^2}\right) + \delta\left(\frac{K_2}{1-\gamma} + \frac{K_1(\overline{R}+\lambda\log|\mathcal{A}|)}{(1-\gamma)^2}\right)\right)$$

$$+ \gamma\mathbb{E}_{x_t,o}^{\xi}\left[\left\|\partial_x Q_{\lambda,x,\xi}^{\pi_{x_t,\xi}^*}(s',a') - \partial_x Q_{\lambda,x,\xi}^{\pi_{x_t,\xi}^o}(s',a')\right\|_{\infty}\right]$$

$$\leq \frac{\gamma}{1-\gamma}\left(|\mathcal{S}|K_1\left(\frac{\lambda\delta\left(|\mathcal{A}||\log l_1| + \frac{2}{l_1}\right)}{1-\gamma} + \frac{\delta|\mathcal{A}|\overline{R}}{(1-\gamma)^2}\right) + \delta\left(\frac{K_2}{1-\gamma} + \frac{K_1(\overline{R}+\lambda\log|\mathcal{A}|)}{(1-\gamma)^2}\right)\right)$$

$$\leq \frac{\delta\gamma}{1-\gamma}\left(|\mathcal{S}|K_1\left(\frac{\lambda\left(|\mathcal{A}||\log l_1| + \frac{2}{l_1}\right)}{1-\gamma} + \frac{|\mathcal{A}|\overline{R}}{(1-\gamma)^2}\right) + \left(\frac{K_2}{1-\gamma} + \frac{K_1(\overline{R}+\lambda\log|\mathcal{A}|)}{(1-\gamma)^2}\right)\right)$$

where we use the intermediate result from before to bound the difference between the value functions.

And so we get that

$$\textbf{(ii)} \leq \frac{2L_f\delta\gamma}{\lambda m(1-\gamma)}\left(|\mathcal{S}|K_1\left(\frac{\lambda\left(|\mathcal{A}||\log l_1| + \frac{2}{l_1}\right)}{1-\gamma} + \frac{|\mathcal{A}|\overline{R}}{(1-\gamma)^2}\right) + \left(\frac{K_2}{1-\gamma} + \frac{K_1(\overline{R}+\lambda\log|\mathcal{A}|)}{(1-\gamma)^2}\right)\right)$$

With that, we can plug everything back together

$$\left\|\mathbb{E}\left[\frac{dF(x_t)}{dx} - \frac{\widehat{dF(x_t)}}{dx}\right]\right\|_{\infty}$$

$$\leq \textbf{(A)} + \textbf{(B)}$$

$$\leq \textbf{(A)} + \textbf{(a)} + \textbf{(b)}$$

$$\leq \textbf{(A)} + \textbf{(a)} + \textbf{(i)} + \textbf{(ii)}$$

$$\leq S_f\delta + \frac{1}{\lambda m}|\mathcal{A}|\delta L_f 2\left(\frac{K_2}{1-\gamma} + \frac{K_1(\overline{R}+\lambda\log|\mathcal{A}|)}{(1-\gamma)^2}\right) + \frac{S_f}{\lambda m}\delta 2\left(\frac{K_2}{1-\gamma} + \frac{K_1(\overline{R}+\lambda\log|\mathcal{A}|)}{(1-\gamma)^2}\right)$$

$$+ \frac{2L_f\delta\gamma}{\lambda m(1-\gamma)}\left(|\mathcal{S}|K_1\left(\frac{\lambda\left(|\mathcal{A}||\log l_1| + \frac{2}{l_1}\right)}{1-\gamma} + \frac{|\mathcal{A}|\overline{R}}{(1-\gamma)^2}\right) + \left(\frac{K_2}{1-\gamma} + \frac{K_1(\overline{R}+\lambda\log|\mathcal{A}|)}{(1-\gamma)^2}\right)\right)$$

$$= \mathcal{O}(\delta)$$

With that we have tackled terms **(1)** and **(2)**. It remains to bound the variance, i.e. term **(3)**

$$\mathbb{E}\left[\left\|\frac{dF(x_t)}{dx} - \frac{\widehat{dF(x_t)}}{dx}\right\|_{\infty}^2\right] \leq 2\left\|\frac{dF(x_t)}{dx} - \mathbb{E}\left[\frac{\widehat{dF(x_t)}}{dx}\right]\right\|_{\infty}^2 + 2\mathbb{E}\left[\left\|\frac{\widehat{dF(x_t)}}{dx} - \mathbb{E}\left[\frac{\widehat{dF(x_t)}}{dx}\right]\right\|_{\infty}^2\right]$$

$$\leq 2\mathcal{O}(\delta^2) + 2\mathbb{E}\left[\left\|\frac{\widehat{dF(x_t)}}{dx} - \mathbb{E}\left[\frac{\widehat{dF(x_t)}}{dx}\right]\right\|_{\infty}^2\right]$$

$$\leq \mathcal{O}(\delta^2) + 2\left(\mathbb{E}\left[\left\|\frac{\widehat{dF(x_t)}}{dx}\right\|_{\infty}^2\right] - \left\|\mathbb{E}\left[\frac{\widehat{dF(x_t)}}{dx}\right]\right\|_{\infty}^2\right)$$

$$\leq \mathcal{O}(\delta^2) + 2\mathbb{E}\left[\left\|\frac{\widehat{dF(x_t)}}{dx}\right\|_{\infty}^2\right]$$

We thus need to bound the second moment.

$$\mathbb{E}\left[\left\|\frac{\widehat{dF(x_t)}}{dx}\right\|_\infty^2\right] \leq \mathbb{E}\left[\left\|\frac{\partial_1 f(x_t, \pi^o_{x_t,\xi}, \xi)}{\partial x} + \frac{1}{\lambda\nu(s)}\frac{\partial_2 f(x_t, \pi^o_{x_t,\xi}, \xi))}{\partial \pi(s,a)}\widehat{\partial_x A^{\pi^o_{x_t,\xi}}_{\lambda,x,\xi}}(s,a)\right\|_\infty^2\right]$$

$$\leq \mathbb{E}\left[\left\|L_f + \frac{1}{\lambda m}L_f\widehat{\partial_x A^{\pi^o_{x_t,\xi}}_{\lambda,x,\xi}}(s,a)\right\|_\infty^2\right]$$

To proceed we upper bound $\widehat{\partial_x A^{\pi^o_{x_t,\xi}}_{\lambda,x,\xi}}(s,a)$ by

$$\widehat{\partial_x A^{\pi^o_{x_t,\xi}}_{\lambda,x,\xi}}(s,a)$$

$$\leq 2\widehat{\partial_x Q^{\pi^o_{x_t,\xi}}_{\lambda,x,\xi}}(s,a)$$

$$\leq \sum_{t=0}^{T_Q}\frac{d}{dx}r(s_t^{\tau_Q}, a_t^{\tau_Q}) + \frac{\gamma}{1-\gamma}\frac{d}{dx}\log P(s^{\tau_Q}_{T_Q+1}; s^{\tau_Q}_{T_Q}, a^{\tau_Q}_{T_Q})\sum_{t=T_Q+1}^{T_Q+T'_Q+1}\gamma^{(t-T_Q-1)/2}\left(r(s_t^{\tau_Q}, a_t^{\tau_Q}) + \lambda H(\pi(\cdot; s_t))\right)$$

$$\leq T_Q K_2 + \frac{\gamma}{1-\gamma}K_1\frac{\overline{R} + \lambda\log|\mathcal{A}|}{1-\gamma^{0.5}}$$

We thus get

$$\mathbb{E}\left[\left\|\frac{\widehat{dF(x_t)}}{dx}\right\|_\infty^2\right] \leq \mathbb{E}_{T_Q}\left[\left\|L_f + \frac{1}{\lambda m}L_f(T_Q K_2 + \frac{\gamma}{1-\gamma}K_1\frac{\overline{R} + \lambda\log|\mathcal{A}|}{1-\gamma^{0.5}})\right\|_\infty^2\right]$$

$$\leq \mathbb{E}_{T_Q}\left[\left\|L_f + \frac{1}{\lambda m}L_f(T_Q K_2 + \frac{\gamma}{1-\gamma}K_1\frac{\overline{R} + \lambda\log|\mathcal{A}|}{1-\gamma^{0.5}})\right\|_\infty^2\right]$$

$$\leq \mathbb{E}_{T_Q}\left[L_f^2\left(1 + 2\frac{1}{\lambda m}T_Q K_2 + 2\frac{1}{\lambda m}\frac{\gamma}{1-\gamma}K_1\frac{\overline{R} + \lambda\log|\mathcal{A}|}{1-\gamma^{0.5}} + \left(\frac{1}{\lambda m}\right)^2(T_Q K_2)^2\right.\right.$$

$$\left.\left. + \left(\frac{1}{\lambda m}\frac{\gamma}{1-\gamma}K_1\frac{\overline{R} + \lambda\log|\mathcal{A}|}{1-\gamma^{0.5}}\right)^2 + 2(\frac{1}{\lambda m})^2\frac{\gamma}{1-\gamma}\frac{T_Q K_2 K_1(\overline{R} + \lambda\log|\mathcal{A}|)}{1-\gamma^{0.5}}\right)\right]$$

$$\leq L_f^2\left(1 + 2\frac{1}{\lambda m}\frac{1}{1-\gamma}K_2 + 2\frac{1}{\lambda m}\frac{\gamma}{1-\gamma}K_1\frac{\overline{R} + \lambda\log|\mathcal{A}|}{1-\gamma^{0.5}} + \left(\frac{1}{\lambda m}\right)^2(K_2)^2\frac{1+\gamma}{(1-\gamma)^2}\right.$$

$$\left. + \left(\frac{1}{\lambda m}\frac{\gamma}{1-\gamma}K_1\frac{\overline{R} + \lambda\log|\mathcal{A}|}{1-\gamma^{0.5}}\right)^2 + 2(\frac{1}{\lambda m})^2\frac{\gamma}{1-\gamma}\frac{K_2 K_1(\overline{R} + \lambda\log|\mathcal{A}|)}{(1-\gamma)(1-\gamma^{0.5})}\right)$$

It is sufficient for our proof that this the second moment is bounded, we denote this constant for now by $C$.

Now we can plug back into (10).

$$\mathbb{E}\left[\left\|\frac{dF(\hat{x}_T)}{dx}\right\|_\infty^2\right] \leq \text{\textcolor{green}{(1)}} + \text{\textcolor{red}{(2)}} + \text{\textcolor{blue}{(3)}}$$

$$\leq \mathcal{O}(\frac{1}{\alpha T}) + \mathcal{O}(\delta) + 2S_f\alpha\left(\mathcal{O}(\delta^2 + C)\right)$$

$$\leq \mathcal{O}(\frac{1}{\alpha T}) + \mathcal{O}(\delta) + 2S_f\alpha\left(\mathcal{O}(\delta^2) + C\right)$$

$$\leq \mathcal{O}(\frac{1}{\alpha T}) + \mathcal{O}(\delta) + \mathcal{O}(\alpha)$$

□

**Proposition C.1.** In this proof we will derive an expression for

$$\frac{df(x, \pi^*_{\lambda,x}, \xi)}{dx}$$

Applying DCT then directly gives the expression for the derivative of $F(x)$. Because we are looking directly at $f$ we can drop any dependence on $\xi$ below to make the proof more readable and concise.

Let

- $p(\mu \to s, t, \pi, x)$ denote the probability under choice $x$ of reaching state $s$ after $t$ steps starting at $\mu$ and following policy $\pi$

- $p(\mu \to s, a, t, \pi, x)$ denote the probability under choice $x$ of reaching state $s$ after $t$ steps and then taking action $a$ starting at $\mu$ and following policy $\pi$

- $p(\mu \to s, a, s', t, \pi, x)$ denote the probability under choice $x$ of reaching state $s'$ after $t$ steps having previously been in state $s$ and having taken action $a$, starting at $\mu$ and following policy $\pi$

Note we drop the dependence on $\xi$ for the proof. Assuming $\overline{V}(s)$ is differentiable for all $s$, we show the following statement by induction.

$$\frac{df(x, \pi^*_{\lambda,x}, \xi)}{dx} = \sum_s \mu_x(s) \frac{d\log\mu_x}{dx}\overline{V}(s) + \sum_{t=1}^{n+1}\sum_s\sum_a\sum_{s'}\gamma^t p(\mu_x \to s, a, s', t, \pi^*_{\lambda,x}, x) d\log P_x(s'; s, a)\overline{V}(s')$$

$$+ \sum_{t=0}^{n}\gamma^t \sum_s\sum_a p(\mu_x \to s, a, t, \pi^*_{\lambda,x}, x)\left(\frac{1}{\lambda}\partial_x A^{\pi^*_{\lambda,x}}_{\lambda,x}\overline{Q}(s,a) + \frac{d\overline{r}_x(s,a)}{dx}\right)$$

$$+ \gamma^{n+1}\sum_s p(\mu_x \to s, t, \pi^*_{\lambda,x}, x)\partial_x\overline{V}(s')$$

Note that taking $n \to \infty$ then directly proves our claim.

**Base case $n = 0$** We proof the statement for $n = 0$.

$$\frac{df(x, \pi^*_x, \xi)}{dx} = \frac{d}{dx}\sum_s \mu(s)\sum_a \pi^*_x(a; s)\overline{Q}(s,a)$$

$$= \sum_s \frac{d\mu_x(s)}{dx}\overline{V}(s) + \sum_s \mu_x(s)\sum_a \pi^*_{\lambda,x}(a; s)\left(\frac{1}{\lambda}\partial_x A^{\pi^*_{\lambda,x}}_{\lambda,x} + \partial_x\overline{Q}(s,a)\right)$$

$$= \sum_s \frac{d\mu_x(s)}{dx}\overline{V}(s) + \sum_s \mu_x(s)\sum_a \pi^*_{\lambda,x}(a; s)\left(\frac{1}{\lambda}\partial_x A^{\pi^*_{\lambda,x}}_{\lambda,x} + \frac{d\overline{r}_x(s,a)}{dx}\right.$$

$$+ \gamma\sum_{s'}\left(P_x(s'; s, a)\frac{d\log P_x(s'; s, a)}{dx}\overline{V}(s') + P_x(s'; s, a)\partial_x\overline{V}(s')\right)\Bigg)$$

$$= \sum_s \mu_x(s)\frac{d\log\mu_x}{dx}\overline{V}(s) + \sum_{t=1}^{1}\sum_s\sum_a\sum_{s'}\gamma^t p(\mu_x \to s, a, s', t, \pi^*_{\lambda,x}, x)d\log P_x(s'; s, a)\overline{V}(s')$$

$$+ \sum_{t=0}^{0}\gamma^t\sum_s\sum_a p(\mu_x \to s, a, t, \pi^*_{\lambda,x}, x)\left(\frac{1}{\lambda}\partial_x A^{\pi^*_{\lambda,x}}_{\lambda,x}\overline{Q}(s,a) + \frac{d\overline{r}_x(s,a)}{dx}\right)$$

$$+ \gamma^1\sum_s p(\mu_x \to s, t, \pi^*_{\lambda,x}, x)\partial_x\overline{V}(s')$$

where we use Proposition E.1 in the second equality.

**Induction step $n \implies n+1$**

$$\sum_s \mu_x(s) \frac{d \log \mu_x}{dx} \overline{V}(s) + \sum_{t=1}^{n+1} \sum_s \sum_a \sum_{s'}' \gamma^t p(\mu_x \to s, a, s', t, \pi^*_{\lambda,x}, x) d \log P_x(s'; s, a) \overline{V}(s')$$

$$+ \sum_{t=0}^n \gamma^t \sum_s \sum_a p(\mu_x \to s, a, t, \pi^*_{\lambda,x}, x) \left( \frac{1}{\lambda} \partial_x A^{\pi^*_{\lambda,x}}_{\lambda,x} \overline{Q}(s, a) + \frac{d\overline{r}_x(s, a)}{dx} \right)$$

$$+ \gamma^{n+1} \sum_s p(\mu_x \to s, t, \pi^*_{\lambda,x}, x) \partial_x \overline{V}(s')$$

$$= \sum_s \mu_x(s) \frac{d \log \mu_x}{dx} \overline{V}(s) + \sum_{t=1}^{n+1} \sum_s \sum_a \sum_{s'}' \gamma^t p(\mu_x \to s, a, s', t, \pi^*_{\lambda,x}, x) d \log P_x(s'; s, a) \overline{V}(s')$$

$$+ \sum_{t=0}^n \gamma^t \sum_s \sum_a p(\mu_x \to s, a, t, \pi^*_{\lambda,x}, x) \left( \frac{1}{\lambda} \partial_x A^{\pi^*_{\lambda,x}}_{\lambda,x} \overline{Q}(s, a) + \frac{d\overline{r}_x(s, a)}{dx} \right)$$

$$+ \gamma^{n+1} \sum_s p(\mu_x \to s, t, \pi^*_{\lambda,x}, x) \sum_a \pi^*_{\lambda,x}(a; s) \left( \frac{1}{\lambda} \partial_x A^{\pi^*_{\lambda,x}}_{\lambda,x} + \frac{d\overline{r}_x(s, a)}{dx} \right.$$

$$\left. + \gamma \sum_{s'} \left( P_x(s'; s, a) \frac{d \log P_x(s'; s, a)}{dx} \overline{V}(s') + P_x(s'; s, a) \frac{d\overline{V}(s')}{dx} \right) \right)$$

$$= \sum_s \mu_x(s) \frac{d \log \mu_x}{dx} \overline{V}(s) + \sum_{t=1}^{n+2} \sum_s \sum_a \sum_{s'}' \gamma^t p(\mu_x \to s, a, s', t, \pi^*_{\lambda,x}, x) d \log P_x(s'; s, a) \overline{V}(s')$$

$$+ \sum_{t=0}^{n+1} \gamma^t \sum_s \sum_a p(\mu_x \to s, a, t, \pi^*_{\lambda,x}, x) \left( \frac{1}{\lambda} \partial_x A^{\pi^*_{\lambda,x}}_{\lambda,x} \overline{Q}(s, a) + \frac{d\overline{r}_x(s, a)}{dx} \right)$$

$$+ \gamma^{n+2} \sum_s p(\mu_x \to s, t, \pi^*_{\lambda,x}, x) \partial_x \overline{V}(s')$$

which proves our claim. $\qquad \square$

**Proposition E.1** (Best response gradient)**.**

$$\frac{d\pi^*_{x,\xi}}{dx} = \frac{1}{\lambda} \pi^*_{x,\xi}(a; s) \frac{dA^{\pi^*_{x,\xi}}_{\lambda,x,\xi}(s, a)}{dx}$$

*Proof.* This result was previoysly shown by (Chen et al., 2022). We give a short proof below.

$$\frac{d\pi^*(a|s)}{\partial x} = \frac{\left[ \exp(Q(a|s)/\lambda) \cdot \frac{\partial_x Q(a|s)}{\lambda} \sum_{a'} \exp(Q(a'|s)/\lambda) - \exp(Q(a|s)/\lambda) \sum_{a'} \exp(Q(a'|s)/\lambda) \frac{\partial_x Q(a'|s)}{\lambda} \right]}{\left( \sum_{a''} \exp(Q(a|s)/\lambda) \right)^2}$$

$$= \pi^*(a|s) \frac{\partial_x Q(a'|s)}{\lambda} - \frac{\pi^*(a|s) \sum_{a'} \exp(Q(a'|s)/\lambda) \frac{\partial_x Q(a'|s)}{\lambda}}{\sum_{a''} \exp(Q(a|s)/\lambda)}$$

$$= \frac{1}{\lambda} \pi^*(a|s) \partial_x Q(a'|s) - \frac{1}{\lambda} \pi^*(a|s) \sum_{a'} \pi^*(a'|s) \partial_x Q(a'|s)$$

$$= \frac{1}{\lambda} \pi^*(a|s) \left[ \partial_x Q(a'|s) - \partial_x \mathbb{E}_{a' \sim \pi^*(\cdot|s)}[Q(a'|s)] \right]$$

$$= \frac{1}{\lambda} \pi^*(a|s) \left( \partial_x Q(a, s) - \partial_x V(s) \right)$$

$$= \frac{1}{\lambda} \pi^*(a|s) \partial_x A(a|s)$$

$\qquad \square$

**Proposition E.2** (Unbiased advantage derivative estimator). *The output $\widehat{\frac{d}{dx}A^{\pi^o_{x,\xi}}_{\lambda,x,\xi}}(s,a)$ of Algorithm 2 is an unbiased estimate of $\frac{d}{dx}A^{\pi^o_{x,\xi}}_{\lambda,x,\xi}(s,a)$, i.e*

$$\mathbb{E}\left[\widehat{\frac{d}{dx}A^{\pi^o_{x,\xi}}_{\lambda,x,\xi}}(s,a)\right] = \frac{d}{dx}A^{\pi^o_{x,\xi}}_{\lambda,x,\xi}(s,a)$$

*Proof.* Note that we drop any dependence on $x, \xi, \pi^o_{x,\xi}$ for notational clarity. We further emphasize that the trick of truncating a rollout after a geomtrically sampled time to obtain unbiased gradients is commonly used in the RL literature for obtaining unbiased estimates of the standard policy gradient.

We will show that the estimator $\widehat{\frac{d}{dx}Q_\lambda}(s,a)$ given by

$$\sum_{t=0}^{T_Q}\frac{d}{dx}r(s_t,a_t) + \frac{\gamma}{1-\gamma}\frac{d}{dx}\log P(s_{T_Q+1}; s_{T_Q}, a_{T_Q})\sum_{t=T_Q+1}^{T_Q+T'_Q+1}\gamma^{(t-T_Q-1)/2}\left(r(s_t,a_t) + \lambda H(\pi(\cdot; s_t))\right)$$

is unbiased. The same argument then holds for $\widehat{\frac{d}{dx}V_\lambda}(s)$ and shows that $\widehat{\frac{d}{dx}A_\lambda}(s,a)$ is unbiased. We start by noting that

$$\mathbb{E}\left[\sum_{t=0}^{T'_Q}\gamma^{(t)/2}\left(r(s_t,a_t) + \lambda H(\pi(\cdot; s_t))\right)\right]$$

$$=\mathbb{E}_{T'_Q}\mathbb{E}^\pi_s\left[\sum_{t=0}^{T'_Q}\gamma^{(t)/2}\left(r(s_t,a_t) + \lambda H(\pi(\cdot; s_t))\right)\right]$$

$$=\mathbb{E}^\pi_s\left[\mathbb{E}_{T'_Q}\sum_{t=0}^{T'_Q}\gamma^{(t)/2}\left(r(s_t,a_t) + \lambda H(\pi(\cdot; s_t))\right)\right]$$

$$=\mathbb{E}^\pi_s\left[\sum_{t=0}^{\infty}\mathbb{E}_{T'_Q}\left[\mathbb{1}_{t\le T'_Q}\right]\gamma^{(t)/2}\left(r(s_t,a_t) + \lambda H(\pi(\cdot; s_t))\right)\right] \quad (11)$$

$$=\mathbb{E}^\pi_s\left[\sum_{t=0}^{\infty}\gamma^{t}\left(r(s_t,a_t) + \lambda H(\pi(\cdot; s_t))\right)\right]$$

$$=V_\lambda(s) \quad (12)$$

Where we use Fubini for Equation (11) and DCT and the fact that $T'_Q \sim \text{Geo}(1-\gamma^{0.5})$ in Equation (12). From the argument above and the fact that $T_Q$ and $T'_Q$ were sampled independently, it immediately follows that

$$\mathbb{E}_{T_Q,T'_Q}\mathbb{E}^\pi_{s,a}\left[\sum_{t=0}^{T_Q}\frac{d}{dx}r(s_t,a_t) + \frac{\gamma}{1-\gamma}\frac{d}{dx}\log P(s_{T_Q+1}; s_{T_Q}, a_{T_Q})\sum_{t=T_Q+1}^{T_Q+T'_Q+1}\gamma^{(t-T_Q-1)/2}\left(r(s_t,a_t) + \lambda H(\pi(\cdot; s_t))\right)\right]$$

$$(13)$$

$$= \mathbb{E}_{T_Q}\mathbb{E}^\pi_{s,a}\left[\underbrace{\sum_{t=0}^{T_Q}\frac{d}{dx}r(s_t,a_t)}_{(1)} + \underbrace{\frac{\gamma}{1-\gamma}\frac{d\log P(s_{T_Q+1}; s_{T_Q}, a_{T_Q})}{dx}V^\pi_\lambda(s_{T_Q})}_{(2)}\right] \quad (14)$$

We will show that the two summands indicated that they form unbiased estimates. Then by linearity of expectation the result

follows. For (1) using Fubini and DCT it holds that

$$
\mathbb{E}_{T_Q}\mathbb{E}_{s,a}^{\pi}\left[\sum_{t=0}^{T_Q}\frac{d}{dx}r(s_t,a_t)\right] = \mathbb{E}_{s,a}^{\pi}\left[\sum_{k=0}^{\infty}(1-\gamma)\gamma^k\sum_{t=0}^{k}\frac{d}{dx}r(s_t,a_t)\right]
$$

$$
= (1-\gamma)\mathbb{E}_{s,a}^{\pi}\left[\sum_{t=0}^{\infty}\sum_{k=t}^{\infty}\gamma^k\frac{d}{dx}r(s_t,a_t)\right]
$$

$$
= (1-\gamma)\mathbb{E}_{s,a}^{\pi}\left[\sum_{t=0}^{\infty}\frac{\gamma^t}{1-\gamma}\frac{d}{dx}r(s_t,a_t)\right]
$$

$$
= \mathbb{E}_{s,a}^{\pi}\left[\sum_{t=0}^{\infty}\gamma^t\frac{d}{dx}r(s_t,a_t)\right]
$$

Similarly for (2)

$$
\mathbb{E}_{T_Q}\mathbb{E}_{s,a}^{\pi}\left[\frac{\gamma}{1-\gamma}\frac{d\log P(s_{T_Q+1};s_{T_Q},a_{T_Q})}{dx}V_\lambda^\pi(s_{T_Q})\right]
$$

$$
= \frac{\gamma}{1-\gamma}\mathbb{E}_{s,a}^{\pi}\left[\sum_{t=0}^{\infty}\mathbb{1}_{t=T_Q}\frac{d\log P(s_{t+1};s_t,a_t)}{dx}V_\lambda^\pi(s_t)\right]
$$

$$
= \mathbb{E}_{s,a}^{\pi}\left[\sum_{t=0}^{\infty}\gamma^{t+1}\frac{d\log P(s_{t+1};s_t,a_t)}{dx}V_\lambda^\pi(s_t)\right]
$$

Thus we have

$$
\mathbb{E}_{T_Q}\mathbb{E}_{s,a}^{\pi}\left[\sum_{t=0}^{T_Q}\frac{d}{dx}r(s_t,a_t) + \frac{\gamma}{1-\gamma}\frac{d\log P(s_{T_Q+1};s_{T_Q},a_{T_Q})}{dx}V_\lambda^\pi(s_{T_Q})\right]
$$

$$
= \mathbb{E}_{s,a}^{\pi}\left[\sum_{t=0}^{\infty}\gamma^t\frac{d}{dx}r(s_t,a_t) + \gamma^{t+1}\frac{d\log P(s_{t+1};s_t,a_t)}{dx}V_\lambda^\pi(s_t)\right]
$$

$$
= \partial_x Q_\lambda^\pi(s,a)
$$

which proves the Proposition. □

**Proposition E.3** (Unbiased gradient estimator for $F$)**.** *The gradient estimator described in Algorithm 5 is unbiased for the given policy $\pi_{x,\xi}^o$.*

*Proof.* We need to show that:

$$
\mathbb{E}\left[\underbrace{\left(\sum_{t=0}^{T}\frac{d}{dx}\overline{r}(s_t,a_t)\right)}_{(1)} + \underbrace{\frac{1}{\lambda(1-\gamma)}\partial_x\widehat{A_{\lambda,x,\xi}^{\pi_{x,\xi}^o}}(s_T,a_T)\sum_{t'=T}^{T+T'}\gamma^{(t-T)/2}\overline{r}(s_{t'},a_{t'})}_{(2)}\right.
$$

$$
\left. + \underbrace{\frac{1}{1-\gamma}\partial_x\log P(s_T,a_{T-1},s_{T-1})\sum_{t'=T}^{T+T'}\gamma^{(t'-T)/2}\overline{r}(s_{t'},a_{t'})}_{(3)}\right]
$$

$$
= \mathbb{E}_\xi\left[\mathbb{E}_{s_0\sim\mu}^{\pi_{x,\xi}^o}\left[\sum_{t=0}^{\infty}\gamma^t\left(\frac{1}{\lambda}\partial_x A_{\lambda,x,\xi}^{\pi_{x,\xi}^o}(s_t,a_t)\overline{Q}(s_t,a_t) + \frac{d}{dx}\overline{r}(s_t,a_t) + \partial_x\log P_{x,\xi}(s_t,a_{t-1},s_{t-1})\overline{V}(s_t)\right)\right]\right]
$$

We can show the claim seperately for (1), (2) and (3). Note for (1) and (3) the claim directly follows from the proof of Proposition E.2. And the proof for (2) works almost identical to the one for (3) in Proposition E.2, relying on the property that a truncation via a geometric distributioin is identical to an infinite trajectory with a discount factor. □

For the next Proposition, consider the following soft Bellmann optimality operator, which has been shown to be a contraction (Dai et al., 2018; Nachum et al., 2017).

$$(\mathcal{T}_\lambda^* V_\lambda)(s) := \lambda \log \left( \sum_{a \in \mathcal{A}} \exp \left( \frac{r(s,a) + \gamma \mathbb{E}_{s'|s,a}[V_\lambda(s')]}{\lambda} \right) \right). \tag{15}$$

Using Equation (15), one can define a standard soft value iteration algorithm (see Algorithm 3 in Appendix D). We show soft value iteration satisfies Assumption 2.4.

**Proposition E.4.** *Algorithm 3 converges, such that* $\left\| \pi_{x,\xi}^* - \pi_{x,\xi}^o \right\|_\infty^2 \leq \delta^2$ *after $T$ iterations, where $T = \mathcal{O}(\log 1/\delta)$.*

*Proof.* From (Mei et al., 2020)[Lemma 24] we have

$$\|\pi^o - \pi^*\|_\infty \leq \|\pi^o - \pi^*\|_1 \leq \frac{1}{\lambda} \left\| Q_\lambda^T - Q_\lambda^* \right\|_\infty$$

Moreover

$$\frac{1}{\lambda} \left\| Q_\lambda^T - Q_\lambda^* \right\|_\infty \leq \frac{1}{\lambda} \left\| V_\lambda^T - V_\lambda^* \right\|_\infty \leq \frac{\gamma^T}{\lambda} \|V_\lambda^*\| \leq \frac{\gamma^T}{\lambda(1-\gamma)} \left( \overline{R} + \lambda \log |\mathcal{A}| \right)$$

where we use the contraction property shown in (Dai et al., 2018; Nachum et al., 2017) and the fact that we instantiate with 0. The claim follows. □

As value iteration assumes knowledge of the transition function and scales badly when the state and action space are large, in practice stochastic methods such as Q-learning are used instead. For this consider the soft Bellman state-action optimality operator (Asadi & Littman, 2017; Haarnoja et al., 2017).

$$(\mathcal{T}_\lambda^* Q_\lambda)(s,a) := r(s,a) + \gamma \mathbb{E}_{s' \sim P(\cdot|s,a)} \left[ \lambda \log \left( \sum_{a' \in \mathcal{A}} \exp \left( \frac{Q_\lambda(s,a')}{\lambda} \right) \right) \right] \tag{16}$$

Correspondingly, we can use Equation (16) to run soft Q-learning, as described in Algorithm 4 in Appendix D. Equivalently to soft value iteration, we can show soft Q-learning satisfies Assumption 2.4.

**Proposition E.5.** *Let $\pi_B$ be sufficiently exploratory, such that the induced Markov chain is ergodic. Then soft Q-learning converges, such that* $\mathbb{E}_o \left[ \left\| \pi_{x,\xi}^* - \pi_{x,\xi}^o \right\|_\infty^2 \right] \leq \delta^2$ *after $T$ iterations, where $T = \mathcal{O}(\frac{\log(1/\delta)}{\delta^2})$.*

*Proof.* We use the following Theorem from (Qu & Wierman, 2020) to prove our claim:

*Theorem (Qu & Wierman, 2020) Let $x \in \mathbb{R}^d$, and $F : \mathbb{R}^d \to \mathbb{R}^d$ be an operator. We use $F_i$ to denote the $i$'th entry of $F$. We consider the following stochastic approximation scheme that keeps updating $x(t) \in \mathbb{R}^d$ starting from $x(0)$ being the all zero vector,*

$$\begin{aligned} x_i(t+1) &= x_i(t) + \alpha_t \left( F_i(x(t)) - x_i(t) + w(t) \right) & \text{for } i = i_t, \\ x_i(t+1) &= x_i(t) & \text{for } i \neq i_t, \end{aligned}$$

*where $i_t \in \{1, \ldots, d\}$ is a stochastic process adapted to a filtration $\mathcal{F}_t$, and $w(t)$ is some noise. Assume the following:*

*Assumption 1 (Contraction) (a) Operator $F$ is $\gamma$ contraction in $\|\cdot\|_\infty$, i.e. for any $x, y \in \mathbb{R}^d$, $\|F(x) - F(y)\|_\infty \leq \gamma \|x - y\|_\infty$. (b) There exists some constant $C > 0$ s.t. $\|F(x)\|_\infty \leq \gamma \|x\|_\infty + C, \forall x \in \mathbb{R}^d$.*

*Assumption 2 (Martingale Difference Sequence) $w(t)$ is $\mathcal{F}_{t+1}$ measurable and satisfies $\mathbb{E}w(t) \mid \mathcal{F}_t = 0$. Further, $|w(t)| \leq \bar{w}$ almost surely for some constant $\bar{w}$.*

*Assumption 3 (Sufficient Exploration) There exists a $\sigma \in (0, 1)$ and positive integer, $\tau$, such that, for any $i \in \mathcal{N}$ and $t \geq \tau, \mathbb{P}(i_t = i \mid \mathcal{F}_{t-\tau}) \geq \sigma$.*

*Suppose Assumptions 1,2 and 3 hold. Further, assume there exists constant $\bar{x} \geq \|x^*\|_\infty$ s.t. $\forall t, \|x(t)\|_\infty \leq \bar{x}$ almost surely. Let the step size be $\alpha_t = \frac{h}{t+t_0}$ with $t_0 \geq \max(4h, \tau)$, and $h \geq \frac{2}{\sigma(1-\gamma)}$. Then, with probability at least $1 - \delta$,*

$$\|x(T) - x^*\|_\infty \leq \frac{12\bar{\epsilon}}{1-\gamma} \sqrt{\frac{(\tau+1)h}{\sigma}} \sqrt{\frac{\log\left(\frac{2(\tau+1)T^2 n}{\delta}\right)}{T+t_0}} + \frac{4}{1-\gamma} \max\left(\frac{16\bar{\epsilon}h\tau}{\sigma}, 2\bar{x}(\tau+t_0)\right) \frac{1}{T+t_0},$$

*where $\bar{\epsilon} = 2\bar{x} + C + \bar{w}$.*

Note that exactly like in the setting above our algorithm can be seen as a stochastic approximation scheme where we update $Q$ asynchronously just like $x$ above in the following way

$$Q_{s_t, a_t}(t+1) = Q_{s_t, a_t}(t) + \alpha_t \left( F_{s_t, a_t}(Q(t)) - Q_{s_t, a_t}(t) + w_t \right)$$
$$Q_{s,a}(t+1) = Q_{s,a}(t) \qquad\qquad\qquad \text{for } s, a \neq s_t, a_t,$$

where

$$F_{s_t, a_t}(Q) = r(s, a) + \gamma \mathbb{E}_{s' \sim P(\cdot | s, a)} \left[ \lambda \log \left( \sum_{a' \in \mathcal{A}} \exp \left( \frac{Q(s, a')}{\lambda} \right) \right) \right]$$

and for the errors:

$$w_t = r(s_t, a_t) + \gamma \lambda \log \left( \sum_{a' \in \mathcal{A}} \exp \left( \frac{Q_\lambda(s_{t+1}, a')}{\lambda} \right) \right)$$
$$- r(s_t, a_t) + \gamma \mathbb{E}_{s' \sim P(\cdot; s_t, a_t)} \left[ \lambda \log \left( \sum_{a' \in \mathcal{A}} \exp \left( \frac{Q_\lambda(s_{t+1}, a')}{\lambda} \right) \right) \right]$$

We now show that $F$ satisfies the assumptions of the Theorem from (Qu & Wierman, 2020) and use the result to prove our own claim.

In the following we let $\mathcal{F}_t$ be the $\sigma$–algebra generated by the random variables $(s_0, a_0, \cdots, s_t, a_t)$

First we state the following identity from (Nachum et al., 2017)

$$T_\lambda^*(Q)(s, a) = r(s, a) + \gamma \mathbb{E}_{s' \sim P(\cdot | s, a)} \left[ \lambda \log \left( \sum_{a' \in \mathcal{A}} \exp \left( \frac{Q(s, a')}{\lambda} \right) \right) \right]$$
$$= r(s, a) + \gamma \mathbb{E}_{s' \sim P(\cdot | s, a)} \left[ \max_\pi \langle Q(\cdot, s'), \pi \rangle + \lambda H(\pi; s') \right].$$

We can use the above to show that $T_\lambda^*$ is a contraction. Indeed we have:

$$\left\| T_\lambda^*(Q_1) - T_\lambda^*(Q_2) \right\|_\infty$$
$$= \left\| r(s, a) + \gamma \max_\pi \sum_{s'} P(s'; s, a) \left( \langle Q_1(\cdot, s'), \pi \rangle + \lambda H(\pi; s') \right) \right.$$
$$\left. - r(s, a) - \gamma \max_\pi \sum_{s'} P(s'; s, a) \left( \langle Q_2(\cdot, s'), \pi \rangle + \lambda H(\pi; s') \right) \right\|_\infty$$
$$\leq \gamma \left\| \max_\pi \sum_{s'} P(s'; s, a) \left( \langle Q_1(\cdot, s'), \pi \rangle - \langle Q_2(\cdot, s'), \pi \rangle \right) \right\|_\infty$$
$$\leq \gamma \left\| Q_1 - Q_2 \right\|_\infty$$

Moreover, it holds that

$$F(Q) \leq \overline{R} + \gamma \left\| Q \right\|_\infty + \lambda \log |\mathcal{A}|$$

So we can set $C = \overline{R} + \lambda \log |\mathcal{A}|$

Next we note that $w_t$ is $\mathcal{F}_{t+1}$–measurable (it depends on $s_{t+1}$) and that

$$\mathbb{E}[w_t | \mathcal{F}_t] = 0$$

Moreover $w(t)$ is bounded by $\overline{w} = \frac{2\gamma(\overline{R} + \lambda \log |\mathcal{A}|)}{1 - \gamma}$

Further have assumed that the behavioural policy $\pi_B$ is sufficiently exploratory. Let $\tilde{\mu}$ be the corresponding stationary distribution, $\mu_{\min} = \inf_{s,a} \tilde{\mu}(s,a)$ and $t_{mix}$ the mixing time. Then (Qu & Wierman, 2020) show that for $\sigma = \frac{1}{2}\mu_{\min}$ and $\tau = \lceil \log_2(\frac{2}{\mu_{\min}}) \rceil t_{mix}$ it holds that

$$\forall s \in \mathcal{S}, a \in \mathcal{A}, \forall t \geq \tau : \mathbb{P}(s_t, a_t = s, a | \mathcal{F}_{t-\tau}) \geq \sigma \tag{17}$$

Moreover, we note that $Q(t)$ and $Q^*$ are bound by $\overline{x} = \frac{\overline{R} + \lambda \log |\mathcal{A}|}{1 - \gamma}$

By plugging into the Theorem from (Qu & Wierman, 2020) we thus have the following result:

Let $\alpha_t = \frac{h}{t + t_0}$ with $t_0 \geq \max\left(4h, \lceil \log_2 \frac{2}{\mu_{\min}} \rceil t_{\mathrm{mix}}\right)$ and $h \geq \frac{4}{\mu_{\min}(1-\gamma)}$. Then, with probability at least $1 - p$,

$$\|Q(T) - Q_\lambda^*\|_\infty \leq$$

$$\leq \frac{60(\overline{R} + \lambda \log |\mathcal{A}|)}{(1-\gamma)^2} \sqrt{\frac{2\left(\lceil \log_2 \frac{2}{\mu_{\min}} \rceil t_{\mathrm{mix}} + 1\right) h}{\mu_{\min}}} \sqrt{\frac{\log\left(\frac{2\left(\lceil \log_2 \frac{2}{\mu_{\min}} \rceil t_{\mathrm{mix}} + 1\right) T^2 |\mathcal{S}||\mathcal{A}|}{p}\right)}{T + t_0}}$$

$$+ \frac{4(\overline{R} + \lambda \log |\mathcal{A}|)}{(1-\gamma)^2} \max\left(\frac{160h \lceil \log_2 \frac{2}{\mu_{\min}} \rceil t_{\mathrm{mix}}}{\mu_{\min}}, 2\left(\lceil \log_2 \frac{2}{\mu_{\min}} \rceil t_{\mathrm{mix}} + t_0\right)\right) \frac{1}{T + t_0}$$

Let us denote the bound above by **(A)**

Let us choose $p = \mathcal{O}(\delta^2)$. With probability $p$ $\|Q(T) - Q_\lambda^*\|_\infty$ is not bounded by the term above. However it is always upper bound by $\frac{2(\overline{R} + \lambda \log(|\mathcal{A}|))}{1 - \gamma}$.

At the same time

$$\textbf{(A)} = \mathcal{O}\left(\sqrt{\log(1/\delta)}\sqrt{1/T}\right)$$

By setting $T = \mathcal{O}(\frac{\log(1/\delta)}{\delta^2})$ and using (Mei et al., 2020)[Lemma 24] we get

$$\mathbb{E}_o\left[\|\pi^o - \pi^*\|_\infty^2\right]$$

$$\leq \left(\frac{1}{\lambda}\right)^2 \mathbb{E}_o\left[\|Q_\lambda^T - Q_\lambda^*\|_\infty^2\right]$$

$$\leq (1-p)\textbf{(A)}^2 + p\left(\frac{2(\overline{R} + \lambda \log(|\mathcal{A}|))}{1 - \gamma}\right)^2$$

$$= \mathcal{O}(\delta^2)$$

$\square$

A popular class of RL algorithms are policy gradient methods such as REINFORCE (Williams, 1992). For the entropy-regularised problem, it generally makes sense to choose a softmax parametrization for the policy (Mei et al., 2020). We defer the details to Algorithm 6 in Appendix D and present the following convergence result, which shows using vanilla policy gradient method for the lower level also fulfills Assumption 2.4—at least asymptotically.

**Proposition E.6.** *Vanilla policy gradient with softmax parameterization converges, such that $\forall \delta, \exists T, \forall t \geq T$ : $\left\|\pi_{x,\xi}^* - \pi_{t,x,\xi}^o\right\| \leq \delta^2$, where $\pi_{t,x,\xi}^o$ is the computed policy after $t$ iterations.*

*Proof.* As in most proofs we drop the subscripts for $x, \xi$. The proof is an adaptation of the one presented for Lemma 16 in (Mei et al., 2020). We denote by $\pi_t$ the iterates of the policies of the algorithm and by $V_\lambda^{\pi_t}(\mu)$ the corresponding value function with starting distribution $\mu$. It can be shown that $V_\lambda^\pi$ is $s$-smooth for some $s$ (Mei et al., 2020). Choosing stepsize $1/s$, we have by sufficient increase that the value functions increase monotonically, i.e.

$$\forall t : V_\lambda^{\pi_{t+1}}(\mu) \geq V_\lambda^{\pi_t}(\mu)$$

At the same time it holds that

$$V_\lambda^{\pi_t}(\mu) \leq \frac{\overline{R} + \lambda \log \mathcal{A}}{1 - \gamma}$$

By monotone convergence it thus follows that $V_\lambda^{\pi_t}(\mu) \to V_\lambda^*(\mu)$, where $V_\lambda^*(\mu)$ is the maximum possible value.

Since $\pi_t \in \Delta(\mathcal{A})^{|\mathcal{S}|}$ and $\Delta(\mathcal{A})^{|\mathcal{S}|}$ is compact it follows that $\{\pi_t\}_t$ has a convergent subsequence $\{\pi_{t_k}\}_k$. Denote by $\pi^*$ the limit of this subsequence. It has to hold that $V_\lambda^{\pi^*}(\mu) = V_\lambda^*(\mu)$.

Now assume that $\{\pi_t\}_t$ does not converge to $\pi^*$. In that case

$$\exists \epsilon, \forall t, \exists t' \geq t : \|\pi^* - \pi_{t'}\|_\infty > \epsilon$$

Note that due to entropy regularization $V_\lambda^*(\mu)$ is the unique maximum. This means that

$$\exists \kappa : \max\{V_\lambda^\pi \mid \|\pi_\lambda - \pi^*\|_\infty \geq \epsilon\} + \kappa < V_\lambda^*$$

It follows then that

$$\forall t, \exists t' \geq t : \left\| V_\lambda^{\pi^*} - V_\lambda^{\pi_{t'}} \right\|_\infty > \kappa$$

which implies $V_\lambda^{\pi_t}(\mu)$ does not converge to $V^*$, a contradiction and thus it follwos that $\pi_t \to \pi_\lambda^*$ □

The asymptotic guarantee of Vanilla Policy Gradient can be improved to non-asymptotic by using Natural Policy Gradient, as introduced by (Kakade, 2001). We restate the following result from (Cen et al., 2022).

**Proposition E.7** (Linear convergence of exact entropy-regularized NPG, (Cen et al., 2022))**.** *For any learning rate* $0 < \eta \leq (1 - \gamma)/\tau$, *the entropy-regularized NPG updates (18) satisfy*

$$\left\| Q_\lambda^\star - Q_\lambda^{(t+1)} \right\|_\infty \leq C_1 \gamma (1 - \eta\lambda)^t$$

$$\left\| \log \pi_\lambda^\star - \log \pi^{(t+1)} \right\|_\infty \leq 2C_1 \lambda^{-1} (1 - \eta\lambda)^t$$

*for all $t \geq 0$, where*

$$C_1 := \left\| Q_\lambda^\star - Q_\lambda^{(0)} \right\|_\infty + 2\lambda \left( 1 - \frac{\eta\lambda}{1 - \gamma} \right) \left\| \log \pi_\lambda^\star - \log \pi^{(0)} \right\|_\infty.$$

**Proposition E.8** (Improved iteration complexity for the follower)**.** *Using vanilla Q-learning vs our accelerated approach we get the following rates.*

|  | Vanilla | Accelerated |
|---|---|---|
| *Bias* | $\mathcal{O}(2^{K/2} \log(2^{K/2}))$ | $\mathcal{O}(2^{K/2} \log(2^{K/2}))$ |
| *Iter. complexity* | $\mathcal{O}(2^K)$ | $\mathcal{O}(K)$ |

*Proof.* Note that the idea and proof strategy of this speed-up have been put forward in the work of (Hu et al., 2024).

*Vanilla Q-learning* Let us start with showing the rates if we run vanilla Q-learning to estimate $\frac{dF(x)}{dx}$. Let $K > 0$ and we run Q-learning until we have a convergence such that By Proposition E.5, if we run Q-learning for $T = 2^K$ iterations, we get

$$\mathbb{E}\left[ \left\| \pi_{x,\xi}^T - \pi_{x,\xi}^* \right\|_\infty \right] = \mathcal{O}(2^{K/2} \log(2^{K/2}))$$

*Monte-Carlo (MC) Q-learning* Let us now turn to our MC approach. Recall instead of letting Q-learning run for a fixed number of $T$ iterations, we instead first sample a random variable $\hat{k}$ from $1, \ldots, K$ with probability

$$p_k = \frac{2^{-k}}{(1 - 2^{-K})}$$

and then use as estimator

$$\frac{d}{dx} F_{t_K}^{ac} = \frac{d}{dx} F_{t_1} + \frac{\frac{d}{dx} F_{t_{\hat{k}+1}} - \frac{d}{dx} F_{t_{\hat{k}}}}{p_{\hat{k}}}$$

where $t_k = 2^k$ achieves a bias of $\mathbb{E}\left\|\pi_{x,\xi}^{t_k} - \pi_{x,\xi}^*\right\|_\infty \leq \mathcal{O}(2^{k/2}\log(2^{k/2}))$.

As we have already shown in the main text it is an unbiased estimator of the gradient estimate if we use vanilla Q-learning. Indeed, we have

$$\frac{d}{dx}F_{t_K} = \frac{d}{dx}F_{t_1} + \sum_{k=1}^{K-1}\frac{d}{dx}F_{t_{k+1}} - \frac{d}{dx}F_{t_k}$$

$$= \frac{d}{dx}F_{t_1} + \sum_k p_k \frac{\frac{d}{dx}F_{t_{\hat{k}+1}} - \frac{d}{dx}F_{t_{\hat{k}}}}{p_k}$$

$$= \frac{d}{dx}F_{t_1} + \mathbb{E}_{\hat{k}\sim p_{\hat{k}}}\left[\frac{\frac{d}{dx}F_{t_{\hat{k}+1}} - \frac{d}{dx}F_{t_{\hat{k}}}}{p_{\hat{k}}}\right].$$

Therefore it directly follows that for any given $K$, $\frac{d}{dx}F_{t_K}^{ac}$ and $\frac{d}{dx}F_{t_K}$ have the same bias of $\mathcal{O}(2^{K/2}\log(2^{K/2}))$.

However, for the expected number of iterations to run the lower-level Q-learning algorithm, we can show a massive improvement. Indeed, we have for the expected number of iterations:

$$\sum_{k=1}^K t_k \frac{2^{-k}}{1-2^{-K}}$$

$$\leq C\sum_{k=1}^K \frac{1}{2^{-k}}\frac{2^{-k}}{1-2^{-K}}$$

$$\leq \sum_{k=1}^K \frac{1}{1-2^{-K}}$$

$$= \mathcal{O}(K)$$

$\square$

# F. Implementation Details

## F.1. Baseline Algorithms

### F.1.1. ADAPTIVE MODEL DESIGN (CHEN ET AL., 2022)

As noted in Section 3, the Adaptive Model Design (AMD) algorithm (Chen et al., 2022) was proposed for the Regularized Markov Design (RMD) problem which is a special case of Bilevel Optimization on Contextual Markov Decision Processes. In particular, when $\Xi$ is a Diriclet distribution BO-CMDP reduces to the RMD problem. To account for this difference, we modify the AMD algorithm (Algorithm 2 in (Chen et al., 2022)) as described in Algorithm 7. We denote the upper-level reward and value functions with the superscript $u$ in the algorithm.

### F.1.2. ZERO-ORDER ALGORITHM

Algorithm 8 defines the zero-order gradient estimation algorithm described in Section 3. We parametrize the perturbation constant to decrease with the number of iterations such as $u_t = \frac{C}{t}$ where $C$ is a positive constant.

## F.2. Four Rooms

### F.2.1. IMPLEMENTATION DETAILS

We parametrize the penalty function $\tilde{r}$ as the softmax transformation of $x \in \mathbb{R}^{d_s+1}$ where the $i$-th entry of $x$ corresponds to the $i$-th cell in the state space $\mathcal{S}$ and the additional dimension $d_s + 1$ is used to allocate the penalties not effective and also excluded from the penalty term received by the leader at the end of each episode. In particular,

$$\tilde{r}(s,a) = -0.2 * \mathrm{softmax}(s;x)$$

---

**Algorithm 7** (Modified) Adaptive Model Design

---

    **Input:** Iterations $T$, Inner iterations: $K$, Learning rate $\alpha$, Regularization $\lambda$, gradient of the pre-learned model $\nabla_x \log P$, gradient of the reward function $\nabla_x r$

    **Initialize** $x_0$, $Q_0$, $\nabla_x Q_0$, and $\tilde{Q}_0$

    **for** $t = 0$ to $T - 1$ **do**

        $\xi \sim \Xi$

        **for** $k = 0$ to $K - 1$ **do**

            $\pi_{x_t,\xi} \leftarrow \exp(\lambda Q_k(s, \cdot))$

            Calculate $V_k, \nabla_{x_t} V_k, V_k^U, \nabla_{x_t} A_k, A_k^u, \tilde{V}_k$

            $Q_{k+1} \leftarrow \mathcal{T}_{r,\gamma}(V_k)$

            $\nabla_{x_t} Q_{k+1} = \mathcal{T}_{\nabla_{x_t} r, \gamma}(\nabla_{x_t} V_k + V_k \nabla_{x_t} \log P)$

            $Q_{k+1}^u = \mathcal{T}_{r_u, \gamma_u}(V_k^u)$

            $\tilde{Q}_{k+1} \leftarrow \mathcal{T}_{\nabla_{x_t} r^u + \lambda A_k^u \nabla_{x_t} A_k}(\tilde{V}_k + V_k^u \nabla_{x_t} \log P)$

        **end for**

        Set $\widehat{\frac{dF}{dx}} = \tilde{V}_K$

        $x_{t+1} \leftarrow x_t + \alpha \widehat{\frac{dF}{dx}}$

        Reinitialize $Q_0 \leftarrow Q_K$, $\nabla_x Q_0 \leftarrow \nabla_x Q_K$, and $\tilde{Q}_0 \leftarrow \tilde{Q}_K$

    **end for**

    **Output:** Optimised parameter $x_T$

---

---

**Algorithm 8** Zero-Order Algorithm

---

    **Input:** Iterations $T$, Learning rate $\alpha$, Regularization $\lambda$

    **Initialize** $x_0$

    **for** $t = 0$ to $T - 1$ **do**

        $\xi \sim \Xi$

        Sample $z \sim N(0, I_{d_x})$

        $\pi_{x_t,\xi}^o \leftarrow \texttt{OraclePolicy}(x_t, \xi)$

        $\pi_{x_t + u_t * z, \xi}^o \leftarrow \texttt{OraclePolicy}(x_t + z u_t, \xi)$

        Set $\widehat{\frac{dF}{dx}} = \frac{f(x + u_t * z, \pi_{x + u_t * z, \xi}^*, \xi) - f(x, \pi_{x,\xi}^*, \xi)}{u_t} z$

        $x_{t+1} \leftarrow x_t + \alpha \widehat{\frac{dF}{dx}}$

    **end for**

    **Output:** $\hat{x}_T \sim U(\{x_0, \ldots, x_{T-1}\})$

---

*Table 1.* Performance over hyperparameters $\beta$ and $\lambda$ for the Four Rooms Problem averaged over 10 random seeds with standard errors. Algorithms perform on-par for most hyperparameters while HPGD outperforms others in few. AMD enjoys low variance due to the non-stochastic gradient updates while Zero-Order suffers from the most variation.

| Parameters | | Algorithms | | |
| --- | --- | --- | --- | --- |
| $\lambda$ | $\beta$ | HPGD | AMD | Zero-Order |
| 0.001 | 1 | **0.91** $\pm$ 0.088 | 0.58 $\pm$ 0.000 | 0.59 $\pm$ 0.059 |
| 0.001 | 3 | 0.51 $\pm$ 0.006 | 0.51 $\pm$ 0.000 | 0.50 $\pm$ 0.005 |
| 0.001 | 5 | 0.46 $\pm$ 0.006 | 0.46 $\pm$ 0.003 | 0.46 $\pm$ 0.007 |
| 0.003 | 1 | 0.95 $\pm$ 0.002 | 1.00 $\pm$ 0.000 | 0.91 $\pm$ 0.048 |
| 0.003 | 3 | **0.73** $\pm$ 0.001 | 0.39 $\pm$ 0.000 | 0.40 $\pm$ 0.028 |
| 0.003 | 5 | 0.29 $\pm$ 0.003 | 0.32 $\pm$ 0.000 | 0.32 $\pm$ 0.002 |
| 0.005 | 1 | 1.17 $\pm$ 0.011 | 1.28 $\pm$ 0.003 | 1.15 $\pm$ 0.026 |
| 0.005 | 3 | 1.01 $\pm$ 0.002 | 1.13 $\pm$ 0.004 | 1.02 $\pm$ 0.027 |
| 0.005 | 5 | 0.87 $\pm$ 0.003 | 0.97 $\pm$ 0.009 | 0.79 $\pm$ 0.027 |

where $\operatorname{softmax}(s; x)$ denotes the value of the softmax transformation of $x$ at the entry corresponding to the state $s$. Note that this parametrization explicitly restricts the maximum available budget for penalties to $-0.2$.

### F.2.2. HYPERPARAMETERS

For the upper-level optimization problem, we use gradient norm clipping of $1.0$. The learning rate for each algorithm has been chosen as the best performing one from $[1.0, 0.5, 0.1, 0.05, 0.01]$ individually. Additionally, we tune the parameter $C$ for the Zero-order algorithm on the values $[0.1, 0.5, 1.0, 2.0, 5.0]$. For Hyper Policy Gradient Descent, we sample $10,000$ environment steps for each gradient calculation.

### F.2.3. HYPERPARAMETER COMPARISON

### F.2.4. ADDITIONAL FIGURES

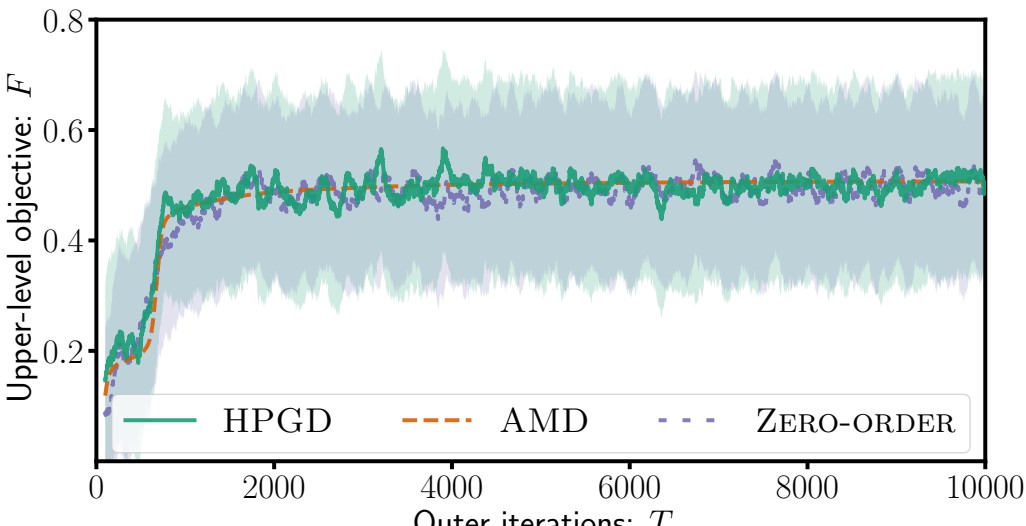

*Figure 3.* Upper-level objective values, $F$, over the number of outer iterations for hyperparameters $\lambda = 0.001$ and $\beta = 3.0$

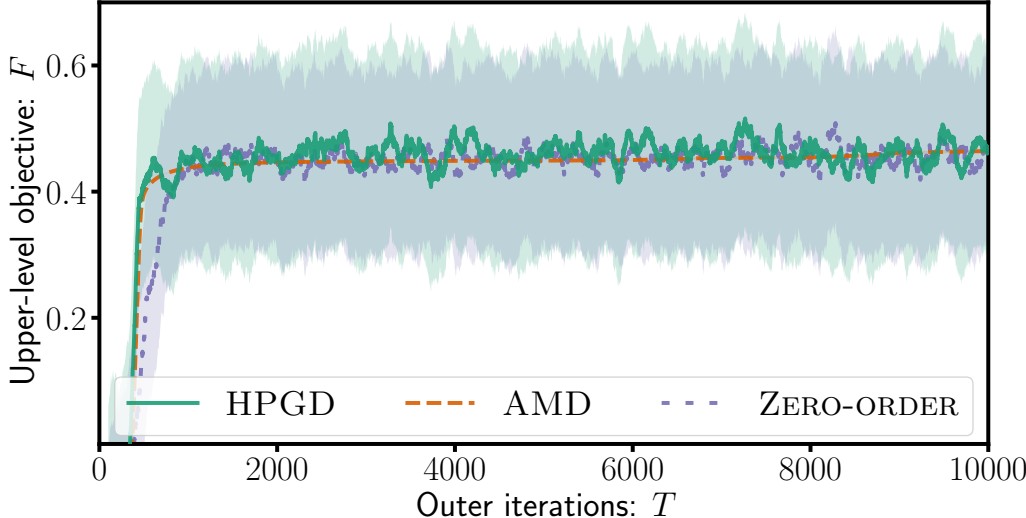

*Figure 4.* Upper-level objective values, $F$, over the number of outer iterations for hyperparameters $\lambda = 0.001$ and $\beta = 5.0$

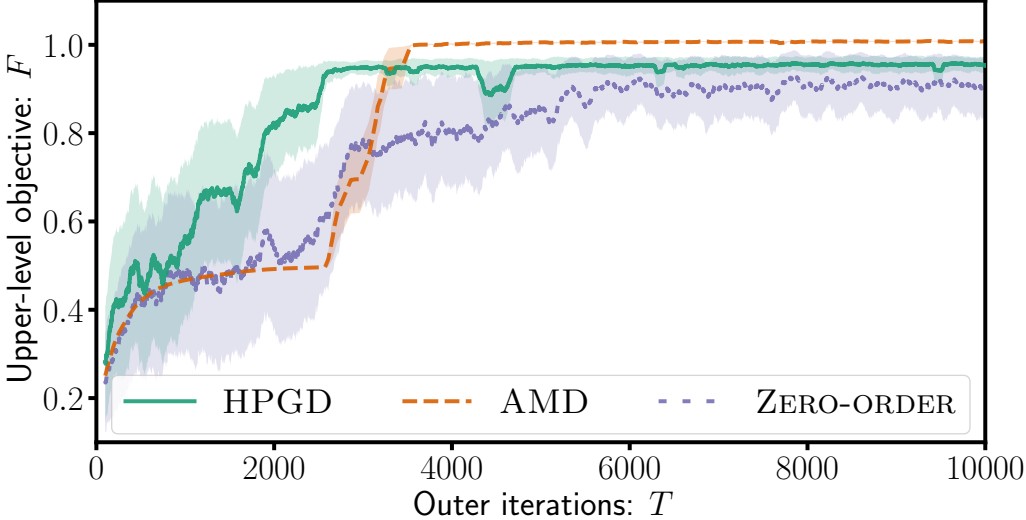

*Figure 5.* Upper-level objective values, $F$, over the number of outer iterations for hyperparameters $\lambda = 0.003$ and $\beta = 1.0$

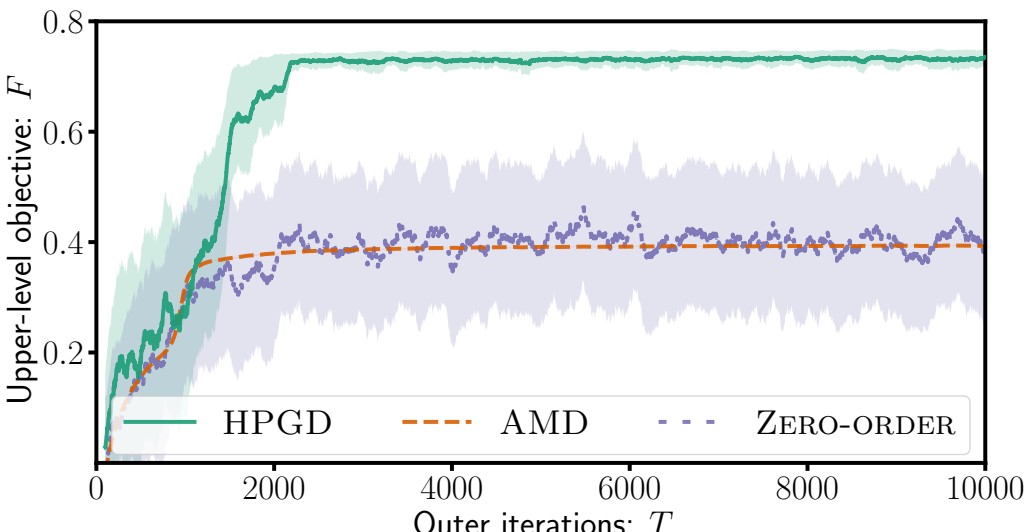

*Figure 6.* Upper-level objective values, $F$, over the number of outer iterations for hyperparameters $\lambda = 0.003$ and $\beta = 3.0$

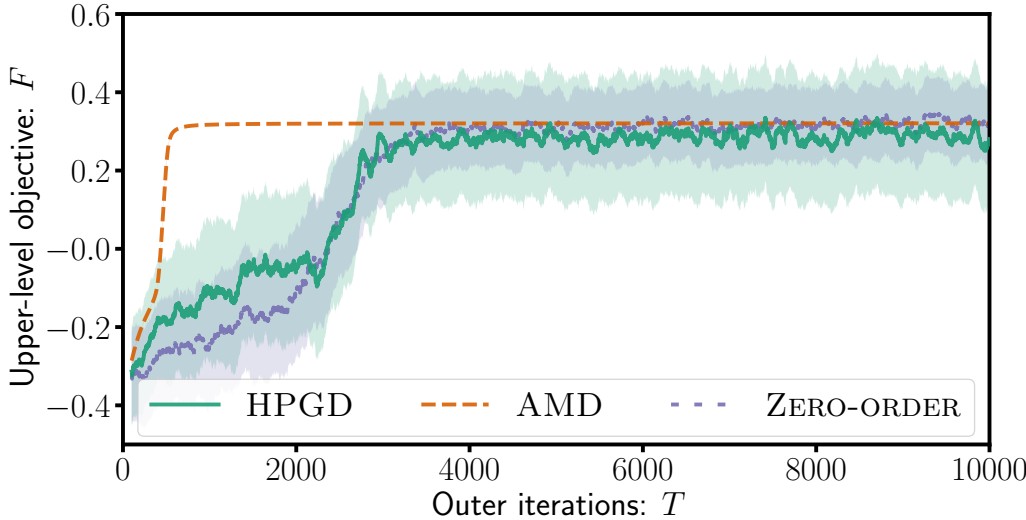

*Figure 7.* Upper-level objective values, $F$, over the number of outer iterations for hyperparameters $\lambda = 0.003$ and $\beta = 5.0$

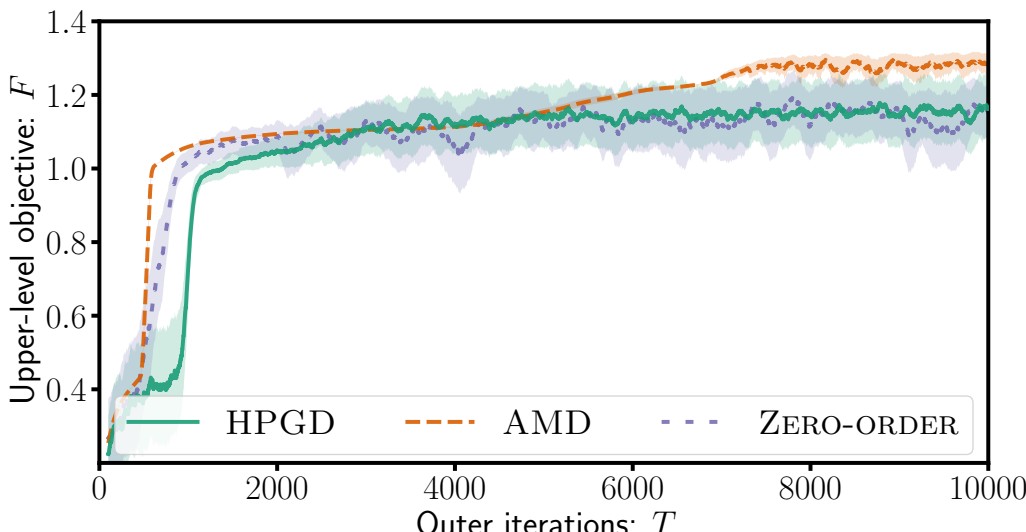

*Figure 8.* Upper-level objective values, $F$, over the number of outer iterations for hyperparameters $\lambda = 0.005$ and $\beta = 1.0$

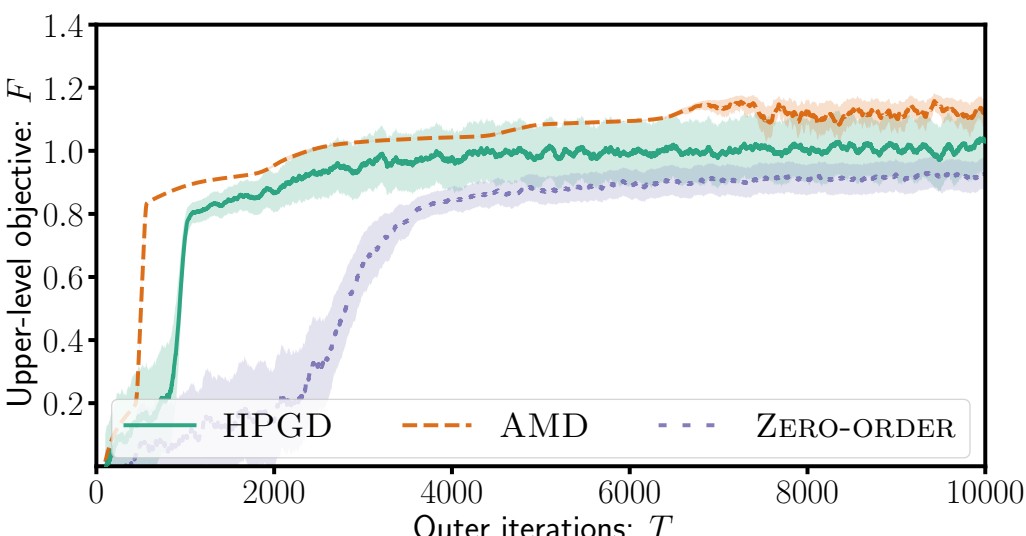

*Figure 9.* Upper-level objective values, $F$, over the number of outer iterations for hyperparameters $\lambda = 0.005$ and $\beta = 3.0$

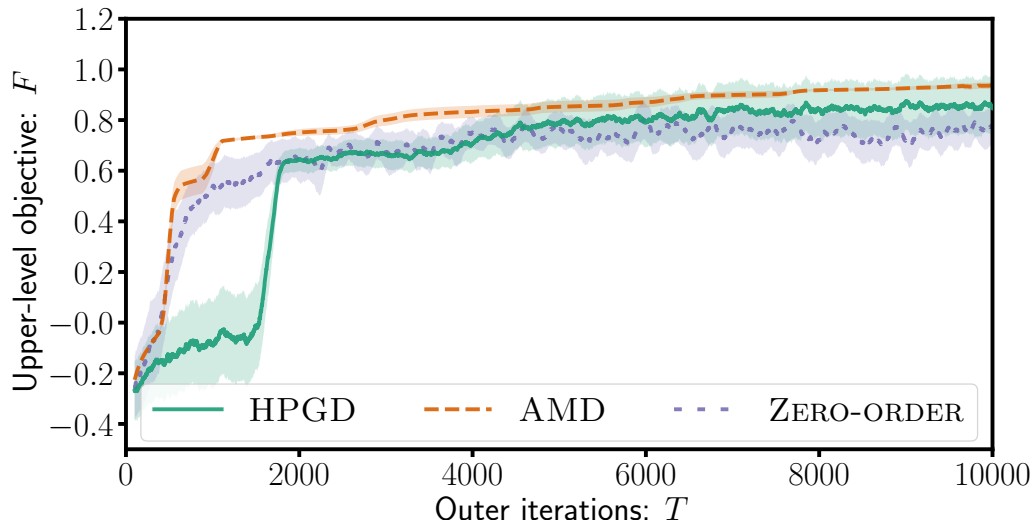

*Figure 10.* Upper-level objective values, $F$, over the number of outer iterations for hyperparameters $\lambda = 0.005$ and $\beta = 5.0$

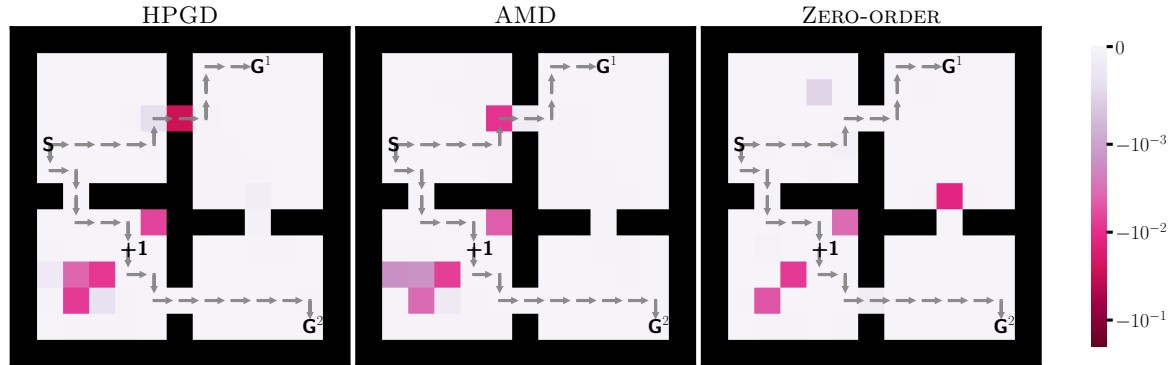

*Figure 11.* Reward penalties given to the lower-level agent in each state of the Four-Rooms problem optimized by the HPGD, AMD, and Zero-Order, respectively, for hyperparameters $\lambda = 0.001$ and $\beta = 3.0$

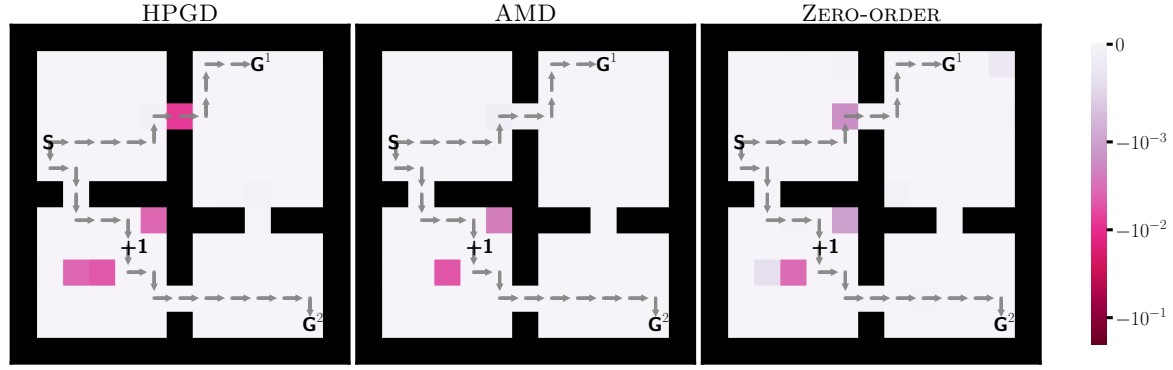

*Figure 12.* Reward penalties given to the lower-level agent in each state of the Four-Rooms problem optimized by the HPGD, AMD, and Zero-Order, respectively, for hyperparameters $\lambda = 0.001$ and $\beta = 5.0$

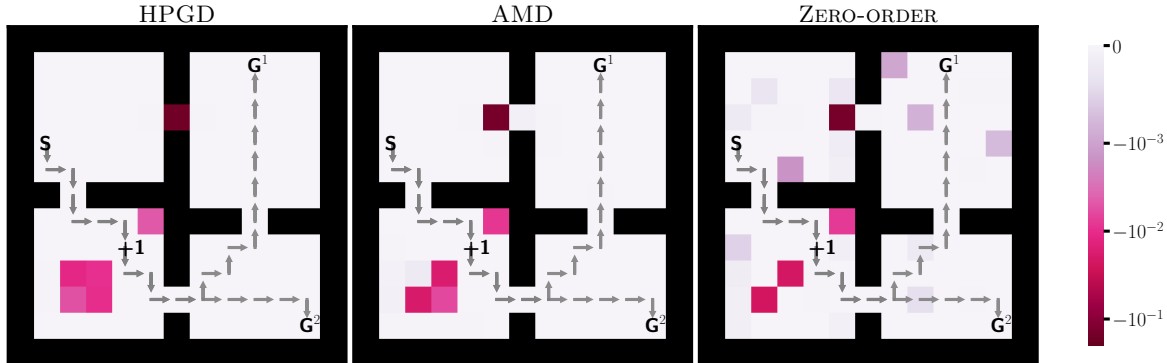

*Figure 13.* Reward penalties given to the lower-level agent in each state of the Four-Rooms problem optimized by the HPGD, AMD, and Zero-Order, respectively, for hyperparameters $\lambda = 0.003$ and $\beta = 1.0$

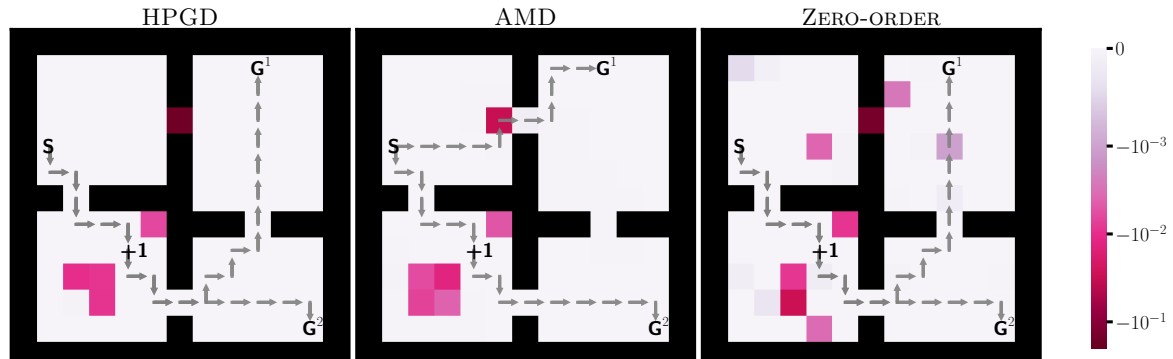

*Figure 14.* Reward penalties given to the lower-level agent in each state of the Four-Rooms problem optimized by the HPGD, AMD, and Zero-Order, respectively, for hyperparameters $\lambda = 0.003$ and $\beta = 3.0$

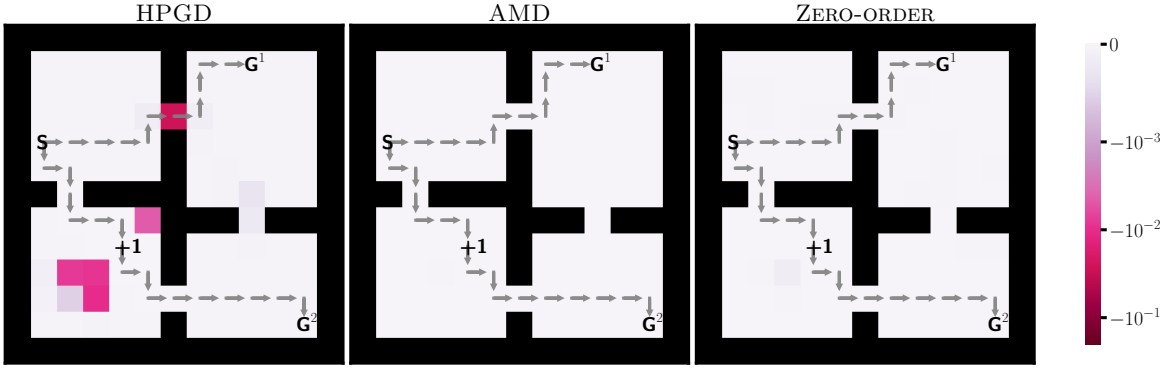

*Figure 15.* Reward penalties given to the lower-level agent in each state of the Four-Rooms problem optimized by the HPGD, AMD, and Zero-Order, respectively, for hyperparameters $\lambda = 0.003$ and $\beta = 5.0$

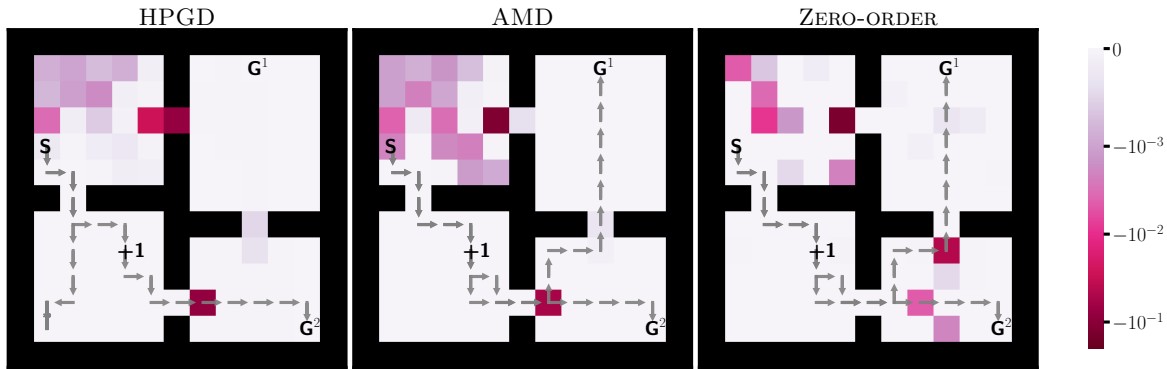

Figure 16. Reward penalties given to the lower-level agent in each state of the Four-Rooms problem optimized by the HPGD, AMD, and Zero-Order, respectively, for hyperparameters $\lambda = 0.005$ and $\beta = 1.0$

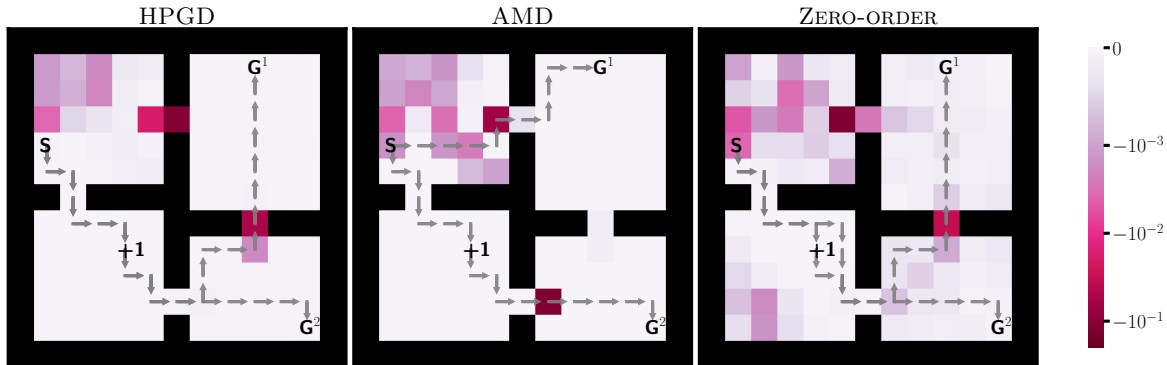

Figure 17. Reward penalties given to the lower-level agent in each state of the Four-Rooms problem optimized by the HPGD, AMD, and Zero-Order, respectively, for hyperparameters $\lambda = 0.005$ and $\beta = 3.0$

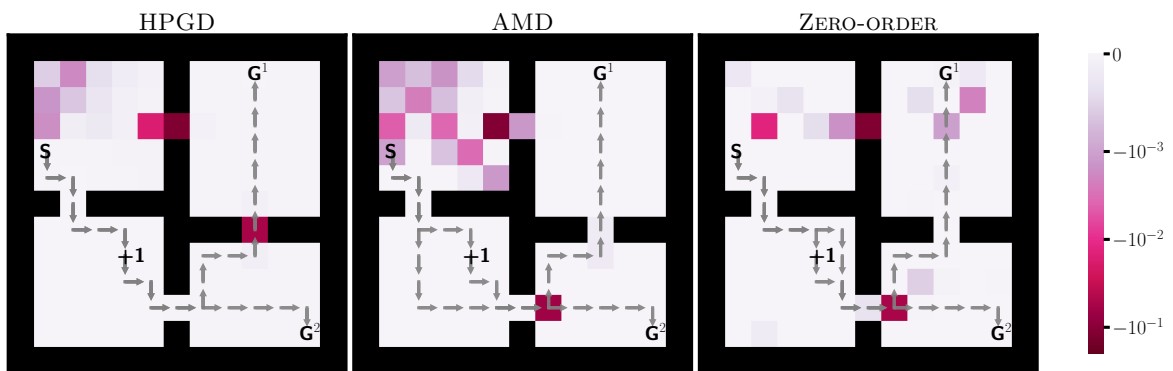

Figure 18. Reward penalties given to the lower-level agent in each state of the Four-Rooms problem optimized by the HPGD, AMD, and Zero-Order, respectively, for hyperparameters $\lambda = 0.005$ and $\beta = 5.0$

### F.3. Computational Costs

We ran our experiments on a shared cluster equipped with various NVIDIA GPUs and AMD EPYC CPUs. Our default configuration for all experiments was a single GPU with 24 GB of memory, 16 CPU cores, and 4 GB of RAM per CPU core. For all parameter configurations reported in Table 1, the total runtime of the experiments for HPGD, AMD, and

Zero-Order were 17, 40, and 2 hours, respectively, totaling 59 hours. Our total computational costs including the intermediate experiments are estimated to be 2-3 times more.

