# OpenReview forum: "Bilevel Optimization with Lower-Level Contextual MDPs"
_ICML.cc/2024/Workshop/Agentic_Markets — Agentic Markets @ ICML'24 Poster_

### Official Review · Reviewer_LSgJ · 2024-06-14
**Strong theoretical results: Leader aims to optimize an objective that depends on followers operating in an environment that the leader can only partially control**

**Rating:** 8
**Confidence:** 4

**Review:**

## Summary
Provides a framework for problems where a leader aims to optimize an objective that depends on followers operating in an environment that the leader can only partially control. The paper introduces Bilevel Optimization on Contextual Markov Decision Processes (BO-CMDP) framework to capture these types of problems. The followers aim to maximize expected reward by finding an optimal CMDP policy and the leader’s objective is to minimize the expected loss function that depends on the follower’s optimal policy. This is a bilevel optimization problem for which the paper proposes the Hyper Policy Gradient Descent Algorithm, where the leader only has access to observed sample trajectories (followers). HPGD converges to a stationary point of the leader’s objective at a rate of O(1/√T).

## Strengths
- The BO-CMDP set-up is novel and very useful for describing and solving the problem defined in the paper. Combining bilevel optimization with contextual MDPs uniquely fits a lot of settings described in the paper (although a more empirical section about applicability would be very useful).
- HPGD to solve BO-CMDP problems works when the leader does not have direct access to the followers’ policies, and with just sample trajectories, is a strong theoretical solution to the problem.
- Extensive proofs of convergence and theoretical analysis are very rigorous and add significantly to the paper.
- The example (toy gridworld environment), although simple, is quite important and well presented.

## Weaknesses
- The paper, despite citing uses of BO-CMDP’s in economics, finance, Meta-RL, etc. only perform experimental evaluation on a simple gridworld task. It does not demonstrate the scalability or applicability of this work to real world tasks.
- HPGD, while theoretically converging to a stationary point of the leader’s objective function, may end up in a local optima. However, this is a problem with many gradient-based optimization algorithms.
- Assumptions about the leader’s information and corresponding results are insufficiently addressed.